# Invariant Predict-and-Combinatorial Optimization under Distribution Shifts

## Abstract

Machine learning has been well introduced to solve combinatorial optimization (CO) problems over the decade, while most of the work only considers the deterministic setting. Yet in real-world applications, decisions have often to be made in uncertain environments, which is typically reflected by the stochasticity of the coefficients of the problem at hand, considered as a special case of the more general and emerging "predict-and-optimize" (PnO) paradigm in the sense that the prediction and optimization are jointly learned and performed. In this paper, we consider the problem of learning to solve CO in the above uncertain setting and formulate it as "predict-and-combinatorial optimization" (PnCO), particularly in a challenging yet practical out-of-distribution (OOD) setting, where we find that in some cases there is decline in solution quality when a distribution shift occurs between training and testing CO instances. We propose the Invariant Predict-and-Combinatorial Optimization (Inv-PnCO) framework to alleviate this challenge. Inv-PnCO derives a learning objective that reduces the distance of distribution of solutions with the true distribution and uses a regularization term to learn invariant decision-oriented factors that are stable in various environments, thereby enhancing the generalizability of predictions and subsequent optimizations. We also provide a theoretical analysis of how the proposed loss reduces the OOD error on decision quality. Empirical evaluation across three distinct tasks on knapsack, visual shortest path planning, and traveling salesman problem covering array, image, and graph input underscores the efficacy of Inv-PnCO to enhance the generalizability, both for predict-then-optimize and predict-and-optimize approaches.

## 1 Introduction

Optimization, especially combinatorial optimization, covers diverse and important applications in the real world, such as supply chain management [12], path planning [56], resource allocation [29], etc. However, many optimizations involve uncertain parameters; for instance in the shortest path problem, the real traveling time on each path could be unknown in advance. Such scenarios call for effective predictions [6] to complete the optimization formulation before the solving procedure, and the adoption of machine learning [42] emerges as a promising direction for decision-making under uncertainty.

Addressing optimizations with unknown coefficients (specifically combinatorial optimization (CO) as the primary focus in this work) is currently approached through two main strategies: "predict-then-optimize" (PtO) and "predict-and-optimize" (PnO, mainly focusing on PnCO for CO problems in the following). PtO [6](or referred to as the "two-stage" approach), as a basic solution, forecasts optimization coefficients using a predictive model supervised by coefficient labels, then employs standard solvers to derive solutions at the test time, while PnO [15; 42; 16](or "decision-focused learning" [62; 60; 43]) train the prediction model oriented towards the ultimate decision objectives with designed surrogate loss. By aligning the prediction goal with the optimization goal in the end-to-end training, PnO is expected to achieve more appropriate error trade-offs [8] and obtain better final decision quality. Recent work [42; 66; 22; 43] on PnO also validates its ability to reduce regret, where regret measures the quality of decisions under uncertainty by comparing to decisions under full information optimization.

However, similar to observations in machine learning tasks [40; 64; 73], models for CO under uncertainty also may exhibit sensitivity to distribution shifts during training and testing stages, and manifest performance degradation when confronted with new environments for both PtO and PnO

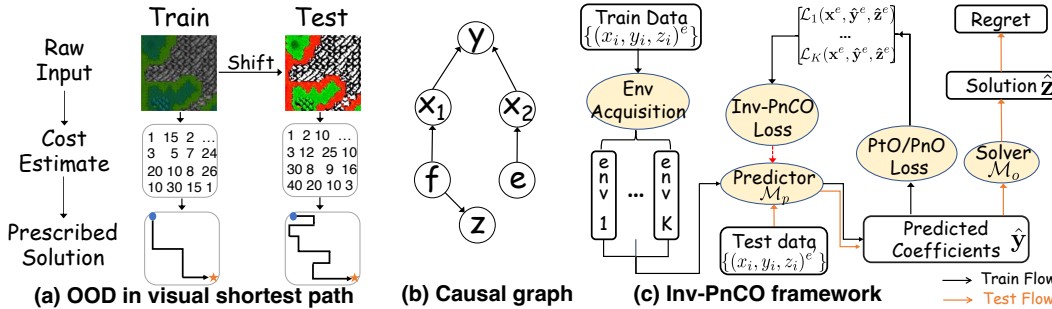

Figure 1: (a) A motivating example: impacts of distribution shift in the visual shortest path (SP) problem using Warcraft dataset. The costs of paths are obtained through predictions by images. Shifts in perceptual mechanisms may lead to inaccurate predictions and sub-optimal decisions. (b) Inter-variable dependencies of perceptual shifts in SP, where notations are listed in the example of SP of Sec 1. (c) Inv-PnCO: a plug-in framework for predict-and-combinatorial optimization by acquiring K environments of data of diverse distributions and then training by the Inv-PnCO loss, a weighted combination of mean and variance terms, to learn invariant PnCO models for improved decision generalizability.

paradigms. Such occurrences are widespread in practical scenarios. For example, the evolution of topological distributions of cities (see Fig. 2) may result in degraded solution quality for Traveling Salesman Problem (TSP) instances [31; 34], especially under uncertain traveling costs [58; 68]. Similarly, in visual shortest path planning [47] in Fig. 1(a), external variables such as weather conditions, variations in lighting, and changes in imaging equipment have the potential to induce shifts in image distributions. The deployment of a trained model on a specific distribution may consequently lead to inaccuracies in cost predictions and yield impractical paths in out-of-distribution (OOD) instances, thereby potentially causing degraded solutions (delays in deliveries to critical clients, etc.).

Various generalizable approaches have been proposed in pure machine learning tasks [40; 64; 67] and some CO tasks [21; 31; 38] to address distribution shifts. However, as shown in Table 1, these methods are not directly applicable to PnCO generalization for two reasons: (1) Similar to that in the independent and identically distributed (IID) setting [15], robust prediction does not always lead to robust decisions in the out-of-distribution optimization under uncertainty, as demonstrated in our experiments. Existing pure ML-based approaches are, therefore, insufficient in this context. (2) No theoretical framework has been investigated to make robust decisions with uncertain coefficients under distribution shifts.

Then, we use an example to demonstrate the challenges of generalization in PnCO and our motivation in a real-world scenario. Fig. 1(a) illustrates an instance of distribution shifts of coefficients in the visual shortest path problem: Decision makers forecast travel costs between grids based on visual images and subsequently determine the route from the upper left to the lower right. However, several factors (denoted as the perceptual mechanism), such as variations in sunlight exposure, weather conditions including clouds, rain, and fog, and imaging devices/parameters [45] including saturation, hue, and contrast, can introduce variability in image distributions. Deploying trained models by independent and identically distributed (IID) data may lead to inaccurate predictions and suboptimal decisions, as evidenced by the performance deterioration observed in the experiments in Table 4.

Hopefully, a key insight from this example toward generalizable decisions lies in identifying invariant decision-oriented factors. As shown in the variable dependence relationship of Fig 1(b), despite variations in perceptual mechanisms (denoted environment $\mathbf{e}$) leading to shifts in the appearances of images (spurious features $\mathbf{x}_2$), terrain serves as a decisive factor (denoted as invariant factor $\mathbf{f}$) influencing both the imagery ($\mathbf{x}_1$, the textures and contours of terrains in images) and the determination of shortest paths (i.e. solution $\mathbf{z}$ of CO problems). On terrain with gentle slopes, the incurred costs are lower, concurrently exhibiting characteristics of flatness in the visual images. These factors remain unaffected by other spurious features $\mathbf{x}_2$. We name $\mathbf{f}$ the invariant decision-oriented factors.

Therefore, we devise a training framework named Invariant Predict-and-Combinatorial Optimization (**Inv-PnCO**) to mitigate the solution degradation caused by distribution shift, which seamlessly

plugs in the current PtO and PnO models. The key advancement of this work is the design of an invariant PnCO framework that captures invariant decision-oriented factors that are stable for the ultimate solutions in various environments. Inv-PnCO proposes a learning objective that ensures the derived solutions closely approximate the true solution distribution and utilize a regularization term to enable the model to capture the invariant factors of PnCO. Based on Assumption 1 that distribution shifts are generated by different environments, and there exist invariant factors whose decisions remain unchanged across different environments, we then theoretically derive a tractable Inv-PnCO loss function to achieve the above goal comprising mean and variance terms of PnO/PtO losses of various environments. Furthermore, we present theoretical results that Inv-PnCO reduces the test error concerning the distribution of final solutions, and validate the efficacy on multiple CO tasks of various distribution shifts. The contributions are summarized as follows:

- We formulate the challenge of out-of-distribution generalization in predict-and-combinatorial optimization (PnCO), and discern the deterioration in decision quality under the distribution shifts between the training and testing sets.

- We propose a novel approach, **Inv**ariant **P**redict-a**n**d-**C**ombinatorial **O**ptimization (**Inv-PnCO**), to enhance generalizability. Inv-PnCO aims to minimize the divergence between the solution distribution and the true distribution, and uses a regularization term to learn invariant features tailored for downstream optimization. Furthermore, we provide theoretical results of how Inv-PnCO reduces the test OOD error of the final prescribed solutions.

- We conduct extensive experiments on distribution shifts of various combinatorial optimization tasks, including artificial, perceptual, and topological shifts in knapsack, visual shortest path (SP) and traveling salesman problem (TSP) covering the input of the array, images and graphs, illustrating the efficacy of both the conventional predict-then-optimize and the predict-and-optimize method.

## 2 PROBLEM FORMULATION

Throughout this paper, we denote variables in bold lowercase letters (e.g., $\mathbf{x}, \mathbf{y}, \mathbf{z}$) and data samples as lowercase letters (e.g., $x_i, y_i, z_i$). Consider a combinatorial optimization problem under uncertainty formulated as:

$$\min_{\mathbf{z} \in \mathcal{Z}} \quad \mathcal{F}(\mathbf{z}, \mathbf{y}, \boldsymbol{\theta}) \quad \text{s.t. } \mathbf{z} \in \text{Constr}(\boldsymbol{\theta}) , \tag{1}$$

where $\mathcal{F}$ is the known and closed-formed optimization objective function, $\mathbf{z} \in \mathcal{Z}$ is the decision variable, $\mathbf{y}$ and $\boldsymbol{\theta}$ are unknown and known parts of optimization parameters, and $\text{Constr}(\boldsymbol{\theta})$ represents the feasible set where decisions satisfy the constraints parameterized by $\boldsymbol{\theta}$. We assume that the parameters in the constraints are known and fixed. We assume the CO objectives as minimization forms for simplicity, whereas maximization forms can be transformed equivalently. The optimization problem is simplified to a minimization one, whereas the maximization problems can be addressed by taking the negation of the objective function.

Although coefficients $\mathbf{y}$ are unknown, in many circumstances, they could be estimated by a prediction model trained on a historical or pre-collected dataset $\mathcal{D} = \{(x_i, y_i)\}$, where $\mathbf{x}$ denotes relevant raw features. The predictive model is denoted by $\hat{\mathbf{y}} = \mathcal{M}_p(\mathbf{x})$, while the optimization solver is represented as $\hat{\mathbf{z}} = \mathcal{M}_o(\hat{\mathbf{y}})$, collectively constituting the system $\mathcal{M}$. A vanilla approach to solving combinatorial optimizations with uncertain coefficients, dubbed "predict-then-optimize" (PtO), is to minimize only the prediction loss and use predictions for the subsequent optimization.

**Definition 1.** (Prediction Optimal) A PnCO system $\mathcal{M}$ achieves **prediction optimal** if the coefficient predictions $\hat{\boldsymbol{y}}$ induced by prediction model $\mathcal{M}_p$ achieve minimum prediction loss on the dataset $\mathcal{D}$:

$$\min_{\mathcal{M}_p} \quad \mathbb{E}_{(x_i, y_i) \sim \mathcal{D}}[\mathcal{L}_{pred}(\hat{y}_i, y_i)] , \tag{2}$$

where $\mathcal{L}_{pred}$ is a training loss specified by the prediction output, e.g. mean squared error (MSE) for regression tasks. This is also referred to as the **two-stage** approach. In contrast to PtO, we next introduce PnO, which learns prediction enhanced by information from optimizations.

**Definition 2.** (Decision Optimal) A PnCO system $\mathcal{M}$ ($\mathcal{M}_p$ and $\mathcal{M}_o$) achieves **decision optimal** if the prescribed solution $\hat{z}$ induced by $\mathcal{M}_o$ with the predicted coefficients $\hat{\boldsymbol{y}}$ achieves its optimal objective induced by $\mathcal{M}_p$ on dataset $\mathcal{D}$:

$$\min_{\mathcal{M}} \quad \mathbb{E}_{(x_i, y_i) \sim D} \left[ \mathcal{F}\left(\hat{z}_i, y_i, \theta\right) \right] . \tag{3}$$

Table 1: Comparison with previously generalizable models against distribution shifts of various types. Inv-PnCO is focused on generalization for predict-and-optimize.

| Previous work | Problem | Task | Generalizability |
|---|---|---|---|
| Mancini et al. [40]/Wu et al. [64]/Yang et al. [67] | Prediction | Image/node/graph classification | Generalization of pure prediction tasks |
| Fu et al. [21]; Luo et al. [38]/Jiang et al. [31] | Optimization | TSP solving | Generalization of pure optimization tasks |
| Inv-PnCO | Predict-and-Optimize | Knapsack/SP/TSP under uncertainty | Generalization of joint prediction and optimization |

In model training, surrogate loss functions $\mathcal{L}(\mathbf{x}, \mathbf{y}, \mathbf{z}; \boldsymbol{\theta})$ (such as SPO loss in Eq. (32)) are usually used to replace the objective in Eq. (3) since we are not able to optimize Eq. (3) directly. This is often due to the inability to differentiate the decision variable concerning coefficients and the discrete nature of decisions $\mathbf{z}$ in the PnO approaches. Although our Inv-PnCO framework applies to any prediction model and combinatorial solvers, in our implementation, the final solution is obtained by an off-the-shelf solver calls following the common practice in the literature [42; 52]. More details are listed in Appendix C.1.

The final decision quality is generally evaluated by regret as in [42; 66; 22; 43], where lower regret indicates better decision quality of $\mathcal{M}$. Regret is the difference between the value of the optimization objective achieved by the estimated coefficient $\hat{\mathbf{z}}$ and the ground-truth coefficient $\mathbf{z}$, when both are evaluated under the true parameter $\mathbf{y}$.

$$\text{Regret}(\hat{\mathbf{y}}, \mathbf{y}) = |\mathcal{F}(\mathbf{z}, \mathbf{y}, \boldsymbol{\theta}) - \mathcal{F}(\hat{\mathbf{z}}, \mathbf{y}, \boldsymbol{\theta})|, \tag{4}$$

To better measure the generalizability of the decision models on the CO under uncertainty, we specify conditional distribution $p(\mathbf{z}|\mathbf{x})$ as the distribution of decision $\mathbf{z}$ given raw feature $\mathbf{x}$, then the conditional Kullback-Leibler (KL) divergence for any two distributions $p_1$ and $p_2$ is given by:

$$D_{KL}\left(p_1(\mathbf{z}|\mathbf{x}) \| p_2(\mathbf{z}|\mathbf{x})\right) := \mathbb{E}_{(x,z) \sim p_1(\mathbf{z}|\mathbf{x})}\left[\log \frac{p_1(\mathbf{z} = z|\mathbf{x} = x)}{p_2(\mathbf{z} = z|\mathbf{x} = x)}\right] \tag{5}$$

We also specify the distribution of solutions learned by system $\mathcal{M}$ as $q(\mathbf{z}|\mathbf{x}) = \mathbb{E}_{y \sim q(\mathbf{y}|\mathbf{x})}[q(\mathbf{z}|\mathbf{y} = y)]$ where $q(\mathbf{y}|\mathbf{x})$, $q(\mathbf{z}|\mathbf{y})$ are distributions induced by predictor $\mathcal{M}_p$ and solver $\mathcal{M}_o$ respectively.

## 3 RELATED WORK

We compare with existing works abbreviated in Table 1, and more discussions are left in Appendix A.

**Predict-and-optimize for optimization under uncertainty** Plenty of recent studies utilize information on downstream optimization problems to enhance prediction models (dubbed "predict-and-optimize" or "decision-focused learning"), which aims to obtain better decisions than the two-stage (or "predict-then-optimize") approach that solely learns the model from the prediction tasks. An influential work is SPO [15] that proposes subgradient-based surrogate functions for linear optimization problems to replace non-differentiable regret functions, as well as a later extended work SPO-relax [42] for a combinatorial counterpart based on continuous relaxation. Later, a class of approaches is developed to deal with differentiable optimization with quadratic programs [3; 62] and further extended to linear [41] and convex [1] objectives. Some other works propose using linear interpolation [47] or perturbation [5] to approximate the gradient, enabling the differentiability of the optimization problem module. These differentiable components are also used to enhance structured output prediction [30], self-supervised [55] and semi-supervised [54] tasks.

However, these works are usually evaluated on i.i.d data while ignoring the risks of out-of-distribution on test data. In this study, we aim to propose a theoretical framework applicable to both PtO and PnO to enhance generalization. Besides, few methods are suitable for various combinatorial optimization tasks as discrete decisions also block the end-to-end training of PnO. Thus, for our experimental investigation, we select two representative approaches: the two-stage approach for PtO and SPO-relax [42] (short as SPO below) for PnO that are applicable to a range of CO tasks.

**Combinatorial optimization and generalization** There are a few studies that also explore the generalization capabilities of combinatorial optimization solvers. Some consider generalizing the trained neural solver to larger problem sizes [21; 34] or different topological distributions [31; 71] on the TSP or vehicle routing problem (VRP). However, these are **orthogonal to ours** as they are

Table 2: Various distribution shifts on combinatorial optimization tasks. "Probabilistic shift" (adopted from [42; 22]) means the change of probability distributions for coefficients, "Perceptual shift" (from [47; 50] ) refers to changes in perceptual mechanisms that result in transformations of images, and "topological shift" (from [7; 57]) means change of graph topology.

| Shift | Problem | Input type | # Train samples | # Test samples | # Decision Variables |
|---|---|---|---|---|---|
| Probabilistic shift | Knapsack | Array | 400 | 200 | 20˜100 |
| Perceptual shift | Shortest path | Image | 10000 | 1000 | 144 |
| Topological shift | TSP | Graph | 400 | 200 | 20 |

focused on the generalizability of the solver and ignore the challenges of uncertain coefficients. Instead, our work treats the solvers as fixed heuristics in implementation and is more concerned with learning robust decision-oriented predictions.

Besides, though generalization toward OOD has been explored in various domains such as images [40], graphs [64; 63], and moleculars [67], it remains largely unexplored in the context of combinatorial optimization problems, especially under uncertainty. We also note that settings in adversarial PnO [18; 65] are different from ours as they are more concerned about the robustness to adversarial attacks but do not include distribution shifts on the train and test set. To the best of our knowledge, our research constitutes a pioneering endeavor that applies the invariance principle to address OOD distribution shifts of CO problems involving uncertain coefficients.

## 4    METHODOLOGY

The out-of-distribution generalization learning objective on predict-and-combinatorial optimization is:

$$\min_{\mathcal{M}} \max_{e \in \mathcal{E}} \quad \mathbb{E}_{(x,y) \sim p(\mathbf{x}, \mathbf{y} | \mathbf{e}=e)}[\mathcal{F}(\hat{z}, y, \theta)], \tag{6}$$

where $\hat{y} = \mathcal{M}_p(x)$ and $\hat{z} = \mathcal{M}_o(\hat{y})$, $\mathbf{e}$ denotes the environmental variable among all possible environments $\mathcal{E}$. Such an objective is hard to solve since we are not able to obtain possibly infinite environments, particularly the environment during testing. However, under the mild assumption that practitioners have access to data from a limited number of domains (like many practices in generalization in ML tasks [37]), we show that we are able to improve existing PtO and PnO generalizability through decision-oriented loss extrapolation.

### 4.1    INVARIANT ASSUMPTION FOR PREDICT-THEN-COMBINATORIAL OPTIMIZATION

Inspired by the example (introduced in Sec 1) above, we aim to develop a generalizable framework capable of learning invariant decision-invariant factors $\mathbf{f}$, so that $\mathcal{M}$ is immune to changes of spurious features $\mathbf{x}_2$ caused by environmental factors $\mathbf{e}$. The underlying assumption, the invariant assumption for PnCO, is given below.

**Assumption 1.** (Invariant PnCO) Assume that various data distributions are generated by different environments, A PnCO system $\mathcal{M}$ satisfies the invariance assumption if $\mathcal{M}$ is capable of learning the invariant factor $\mathbf{f}$ with respect to the decision variable $\mathbf{z}$, so that $p(\mathbf{z}|\mathbf{f}, \mathbf{e} = e) = p(\mathbf{z}|\mathbf{f})$ hold consistently for prescribed solutions $\mathbf{z}$ across any environment $\mathbf{e}$.

Assumption 1 also assumes the existence of invariant factors, and such factors are irrelevant to data generation environment $e$. Also, different from the invariance of predictions in pure machine learning tasks [36], Assumption 1 pertains to the model's ability to sufficiently represent invariant decision-oriented features. Such factors exist in many decision problems. For instance of portfolio optimization with uncertain stock prices, fundamental characteristics such as financial statements and debt levels generally remain stable despite short-term market fluctuations. We may use these invariant factors to design robust PnCO systems.

### 4.2    INVARIANT PREDICT-AND-COMBINATORIAL OPTIMIZATION (INV-PNCO) FRAMEWORK

**Invariant PnCO Training Approach** Since optimizing Eq (6) is intractable when we are not aware of the distribution of test data, achieving a system $\mathcal{M}$ that obtains invariant decisions against distribution

shifts is challenging. Therefore, we introduce a general objective to guide the solutions produced by $\mathcal{M}$ to align with the distribution of optimal decisions of real-world data, and also satisfy the aforementioned invariant decision among environments mentioned in Assumption 1:

$$\min_{\mathcal{M}} \quad D_{KL}(p(\mathbf{z}|\mathbf{x})\|q(\mathbf{z}|\mathbf{y})) + \lambda R(q(\mathbf{z}|\mathbf{x})) \tag{7}$$

where the first term reduces the discrepancy between optimal solution distribution $p(\mathbf{z}|\mathbf{x})$ and distribution $q(\mathbf{z}|\mathbf{y})$ induced by $\mathcal{M}$, which inherently aligns with the goal of decision-oriented predict-and-optimize; the second term $R(\cdot)$ is a regularization that acts on $q(\mathbf{z}|\mathbf{x})$ that ensures $\mathcal{M}$ learns invariant decision-oriented factors. This learning objective could plug in any existing PtO and PnO models.

**Design of Regularization** The subsequent challenge lies in designing a regularization $R(q(\mathbf{z}|\mathbf{x}))$ that ensures $\mathcal{M}$ satisfies the invariant PnCO in Assumption 1, and we proceed with theoretical views.

Let us assume that the training distribution is drawn from the joint distribution $p(\mathbf{x}, \mathbf{z}|\mathbf{e} = e)$, and the test distribution is drawn from $p(\mathbf{x}, \mathbf{z}|\mathbf{e} = e')$. Utilizing the conditional distribution of the solution $\mathbf{z}$ given the raw feature $\mathbf{x}$, the error during training and testing could be represented as $D_{KL}(p_e(\mathbf{z}|\mathbf{x})\|q(\mathbf{z}|\mathbf{x}))$ and $D_{KL}(p_{e'}(\mathbf{z}|\mathbf{x})\|q(\mathbf{z}|\mathbf{x}))$, respectively. In the following, we measure the OOD test decision error under environment $\mathbf{e} = e'$ trained by the proposed Inv-PnCO from an information-theoretic perspective [19]:

**Theorem 1.** *For training data generated by environment $\mathbf{e}$ and any test data generated from environment $\mathbf{e}'$, Eq. (7) with regularization term $R(q(\mathbf{z}|\mathbf{x})) = I_{\mathbf{e},\mathbf{q}}(\mathbf{z}; \mathbf{e}|\mathbf{y})$ upper-bounds KL-divergence $D_{KL}(p_{e'}(\mathbf{z}|\mathbf{x})\|q(\mathbf{z}|\mathbf{x}))$ between the prescribed solution distribution $q(\mathbf{z}|\mathbf{x})$ by model $\mathcal{M}$ and optimal solution distribution $p_{e'}(z|x)$ on condition of $I_{\mathbf{e}',q}(\mathbf{x}; \mathbf{z}|\mathbf{y}) = I_{e,q}(\mathbf{x}; \mathbf{z}|\mathbf{y})$.*

where in the condition, $I_{\mathbf{e},q}(\mathbf{x}; \mathbf{z}|\mathbf{y}) = D_{KL}(q(\mathbf{z}|\mathbf{x}, \mathbf{y})\|q(\mathbf{z}|\mathbf{y}))$ is the mutual information between the raw feature $\mathbf{x}$ and solution $\mathbf{z}$ (produced by the model $\mathcal{M}$ with the distribution of $q(\mathbf{z}|\mathbf{x})$) given coefficient prediction y under environment $\mathbf{e}$, and $I(\mathbf{z}; \mathbf{e}|\mathbf{y}) = D_{KL}(q(\mathbf{z}|\mathbf{y}, \mathbf{e})\|q(\mathbf{z}|\mathbf{y}))$. It is noteworthy that while the optimization solvers are treated as black-box tools in our experiments, Theorem. 1 applies to the entire system $\mathcal{M}$, encompassing both prediction and optimization. The condition in Theorem. 1 can be satisfied when minimizing $D_{KL}(p(\mathbf{z}|\mathbf{x})\|q(\mathbf{z}|\mathbf{y}))$ in the objective (7).

Therefore, Theorem. 1 provides the guidelines for formulating the regularization term. Accordingly, we specify $R(q(\mathbf{z}|\mathbf{x}))$ as $I_e(\mathbf{z}; \mathbf{e}|\mathbf{y})$ to enforce $\mathcal{M}$ learn representations that capture stable decisions across environmental factor $e$. Also, we have proven that minimizing Eq. (7) can reduce the OOD error in the out-of-distribution generalization of the prescribed solution by $\mathcal{M}$. Since this objective can reduce the generalization error of any test environment $\mathbf{e}'$, it equivalently addresses the OOD generalization objective (6) for decision-making.

**Tractable Learning Loss** After resolving the choice of $R(q(\mathbf{z}|\mathbf{x}))$, the difficulty during training is how to make tractable training to minimize $I_e(\mathbf{z}; \mathbf{e}|\mathbf{y})$ with only observable data at hand. Therefore, we propose a tractable estimate that minimizes the above objective and completes the model training.

**Proposition 1.** *Under the invariant condition specified in Assumption 1, the objective in Eq (8) serves as an upper bound for the objective of Eq (7). Specifically, we have:*

$$\min_{\mathcal{M}} \quad \mathrm{Var}_{e\sim\mathcal{E}_{tr}}\left[\mathcal{L}\left(\mathbf{x}^e, \hat{\mathbf{y}}^e, \hat{\mathbf{z}}^e; \boldsymbol{\theta}\right)\right] + \beta\mathbb{E}_{e\sim\mathcal{E}_{tr}}\left[\mathcal{L}\left(\mathbf{x}^e, \hat{\mathbf{y}}^e, \hat{\mathbf{z}}^e; \boldsymbol{\theta}\right)\right]$$

where the superscription $^e$ means corresponding variable under environment $e$. The above loss function is named Inv-PnCO loss, where $\mathrm{Var}(\cdot)$ denotes the variance of losses across training environments $\mathcal{E}_{tr}$, and $\beta$ is a hyper-parameter controlling the balance of two terms, $\mathcal{L}(\cdot)$ is the surrogate loss function for PnO or prediction loss for PtO, specifically we adopt SPO loss as following:

$$\mathcal{L}_{spo}(\mathbf{x}, \mathbf{y}, \mathbf{z}, \hat{\mathbf{y}}, \hat{\mathbf{z}}) = -\mathcal{F}(\tilde{\hat{\mathbf{z}}}, 2\hat{\mathbf{y}} - \mathbf{y}) + 2\mathcal{F}(\mathbf{z}, \hat{\mathbf{y}}) - \mathcal{F}(\mathbf{z}, \mathbf{y}) . \tag{8}$$

where $\mathbf{z}$ denotes the optimal solution using the ground-truth coefficient $\mathbf{y}$, and $\tilde{\hat{\mathbf{z}}}$ denotes solution obtained with the coefficient $(2\hat{\mathbf{y}} - \mathbf{y})$. Intuitively, the first term corresponds to minimizes the discrepancy of decision qualities $p(\mathbf{z}|\mathbf{e}, \mathbf{y})$ for the predictions $\mathbf{y}$ across environments in $\mathcal{E}_{tr}$, while the second term maximizes predictive information and aligns the true solutions with induced solutions by $\mathcal{M}$ of training environments.

**Acquisition of Training Environments** We assume access to data from multiple training domains $\mathcal{E}_{tr}$ in accordance with previous works [37], then data $\mathcal{D}_e = \{(x^e, y^e, z^e)\}$ including raw feature

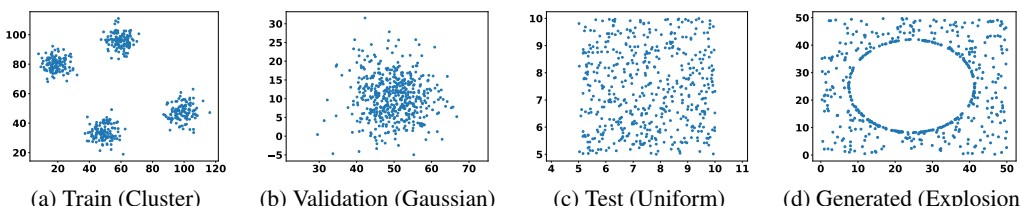

| (a) Train (Cluster) | (b) Validation (Gaussian) | (c) Test (Uniform) | (d) Generated (Explosion) |

Figure 2: Visualization of topological distributions in TSP with unknown costs. Changes in graph topology lead to degradation in the optimization decision quality under unknown coefficients.

$x^e$, coefficients $y^e$ and solutions $z^e$ can be obtained for $K$ different environment $e \in \mathcal{E}_{tr}$, where the generation method is tailored to each optimization task and specified in Appendix C.2 for our implementations.

*Remark* (Heterogeneity of Inv-PnCO environments) Besides the capability of $\mathcal{M}$ to learn invariant decision-oriented factors, the diversity of the acquired environments may be crucial for practical performance. Insufficient diversity in $\mathcal{E}_{tr}$ or direct correlations between environmental factors and targets could undermine the efficacy of Inv-PnCO.

In summary, Inv-PnCO workflow is illustrated in Fig 1(c). For the training, we acquire environments of multiple distributions, and then obtain PtO/PnO losses for each environment. The model is trained by Inv-PnCO loss in Eq. (8) to update the predictor $\mathcal{M}_p$. During testing, the optimization coefficients are predicted by $\mathcal{M}_p$ and solved by $\mathcal{M}_o$, without incurring additional time or space overhead.

## 5 EXPERIMENTS

### 5.1 DATASETS AND EXPERIMENTAL SETUP

We evaluate the generalizability to new environments under the following optimization tasks and distribution shifts, shown in Table 2. All experiments are carried out on a workstation with Intel® i9-7920X, NVIDIA® RTX 2080, and 128GB RAM.

We use the "two-stage" for the PtO method and "SPO" [42] for the PnO method, where the model details are elaborated in Appendix C.1. In each task, we first present the results under IID settings as a reference, then in the "OOD" setting, compare Inv-PnCO with the baseline method, the vanilla empirical risk minimization (ERM) approach, the supervised learning that directly optimizes the loss on the training data. ERM assumes the train/test data to be IID distributed and does not account for distribution shifts. Note that the test sets are identical for IID and OOD settings for direct comparison.

We grid-search the learning rate across {1e-4, 5e-4, 1e-3, 5e-3, 1e-2, 5e-2} for each model, and for Inv-PnCO, we grid-search the hyper-parameter $\beta$ in {0.5, 1.0, 2.0, 4.0} and the number of environments in {1,2,3,4,5}. All models are trained by 300 epochs from scratch and early stops if the regret on the validation set has not improved for 50 epochs. The final result is evaluated on the epoch with the lowest validation regret. Other details are listed in Appendix C, and the code will be released after publication.

### 5.2 KNAPSACK PROBLEM WITH UNKNOWN PROFITS

The **Optimization** procedure aims to maximize the cumulative value of items contained within the knapsack, subject to a capacity constraint, expressed as an integer linear objective function:

$$\mathbf{z}^\star(y) = \arg\max_{\mathbf{z}} \ \Sigma_{i=1}^N \ \mathbf{y}^i \mathbf{z}^i \quad \text{s.t.} \ \Sigma_{i=1}^N \ \mathbf{w}^i \mathbf{z}^i \leqslant C , \tag{9}$$

where the profits $\mathbf{y}^i$ for each item is unknown, and the weights $\mathbf{w}$ are known and identical across different environments. The **Prediction** aims to forecast profits $\mathbf{y}^j$ of the $j$-th item based on the raw feature vector $\mathbf{x}^j$ for each of the $N$ items. The problem is adopted from [42; 22], and the datasets $\mathcal{D} = \{(x_i, y_i)\}$ is generated following previous literature [15]. We evaluate the knapsack with 20 items (and up to 100 in Fig. 9). We use a 3-layer multi-layer perceptron (MLP) as the prediction model and commercial solver Gurobi [23] for optimization. For experiments, the uncertain profits are generated by Gaussian distribution with different mean and variance; thus, probability distribution shifts occur among the training, validation, and test sets. All dataset details are elaborated in Appendix C.2.1.

Table 3: Generalization results for knapsack with unknown profits.

| | IID | | OOD: ERM | | OOD: Inv-PnCO | |
| | Two-stage | SPO | Two-stage | SPO | Two-stage | SPO |
|---|---|---|---|---|---|---|
| Regret | 2.39500 | 2.26000 | 11.22000 | 10.67000 | 9.98500 | 9.10000 |
| Train time | 0.20097 | 1.63326 | 0.21741 | 1.81035 | 0.37711 | 3.68596 |
| Test time | 1.08337 | 0.76298 | 0.95853 | 0.71940 | 0.96731 | 0.74611 |

Table 4: Generalization results for Warcraft shortest path

| | IID | | OOD: ERM | | OOD: Inv-PnCO | |
| | Two-stage | SPO | Two-stage | SPO | Two-stage | SPO |
|---|---|---|---|---|---|---|
| Regret | 11.54528 | 10.80689 | 18.73675 | 13.68741 | 13.5696 | 13.04145 |
| Train time | 0.29022 | 1.78750 | 0.26658 | 1.69342 | 1.39788 | 6.91911 |
| Test time | 0.28751 | 0.30191 | 0.29672 | 0.29138 | 0.39828 | 0.57588 |

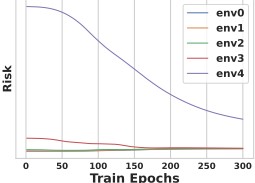 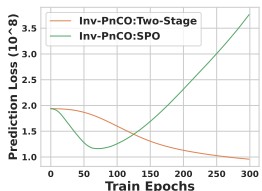 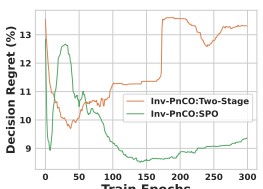

(a) Inv-PnCO-SPO training loss    (b) Training prediction loss    (c) Testing decision regret

Figure 3: Loss curves for each environment, prediction loss, and decision quality (in regret) throughout training/testing of Inv-PnCO framework on the knapsack problem. The PnO approach (SPO) within Inv-PnCO demonstrates better decision-making with lower regret despite exhibiting higher prediction loss, owing to leveraging information from the optimization task. Full images are in Fig. 7.

We present the generalizability results in Table 3. We observe that in the "IID" setting, SPO achieves lower regret than the "two-stage", as it optimizes the surrogate of final decision quality for decision optimal instead of prediction optimal. Further, the out-of-distribution setting "OOD": ERM shows that performance drops significantly for both the PtO approach (two-stage), and the PnO approach (SPO). Lastly, we observe that our results shown in "OOD: Inv-PnCO" significantly reduce the regret compared to ERM for both two-stage and SPO, which validates the improved generalization ability against OOD test data. Besides, we may notice the proposed Inv-PnCO framework does not affect the runtime at the test stage, though it may take affordably more time during the training. Note the runtime variations in testing time stem from machine disturbances and random factors, yet they share an identical procedure that comprises one prediction and one subsequent solver call. The training process is shown in Figure 3, and more in the appendix.

## 5.3 VISUAL SHORTEST PATH PLANNING WITH UNKNOWN COST

The **Optimization** goal is to plan the route with minimum cost on the grid from the upper-left cell to the lower-right cell within the Warcraft terrain map dataset [25] [1]. The agent can control moving to adjacent cells in the grid, where the cost is measured by $N \times N$ cells. The **Prediction** task is to estimate the cost of each grid cell from image input. The task is adopted from [47; 50], and we use ResNet [26] for cost predictions and Dijkstra algorithm [14] as the solver.

Distribution shifts in various perceptual mechanisms frequently occur in the real world. As illustrated in Fig 1(a) and Sec. 4.1, during the acquisition of images, external environmental factors and perceptual characteristics, such as saturation, contrast, and brightness in camera parameters, introduce disparate distributions in the obtained raw images. In this task, we explore how such perceptual shifts affect problem-solving in such a "visual-optimization" task. In our experiments, we conduct different

---

[1]https://github.com/war2/war2edit

Table 5: Generalization results for TSP with unknown costs.

| | IID | | OOD: ERM | | OOD: Inv-PnCO | |
|---|---|---|---|---|---|---|
| | Two-stage | SPO | Two-stage | SPO | Two-stage | SPO |
| Regret | 82.88278 | 33.75459 | 143.32407 | 104.42732 | 100.50798 | 100.35209 |
| Train time | 0.04259 | 2.73619 | 0.01473 | 0.75054 | 0.18914 | 2.90157 |
| Test time | 2.66494 | 1.69280 | 10.11881 | 2.35611 | 2.11336 | 2.0405 |

image transforms on train and validation sets and keep the original image as the test distribution, shown in Fig 5 in the appendix and elaborated in Appendix C.2.2.

Table 4 illustrates that performance in the OOD setting degrades for both the two-stage and SPO approaches compared to the IID setting. When trained with Inv-PnCO, the degradation of regret significantly diminishes due to the Inv-PnCO's ability to learn invariant features across environments, leading to more robust models for both PtO and PnO in response to distribution shifts. Furthermore, lower regret is observed with the PnO method SPO compared to the two-stage approach across IID and OOD settings for both ERM and Inv-PnCO, demonstrating the advantage of decision-focused learning over prediction-oriented to achieve the decision-optimal, as well as its better inherent robustness to distribution shifts. We also observe that under the OOD setting, results of Inv-PnCO for SPO are comparable to those of the two-stage approach. This may indicate the inherent difficulty in achieving robust solutions for complex optimization tasks. Similar to the knapsack task, Inv-PnCO framework maintains an affordable increase in training time without incurring additional test time.

## 5.4 TRAVELLING SALESMAN PROBLEM WITH UNKNOWN COSTS

Suppose a few cities are fully connected and represented in a graph. The goal of TSP is to determine a sequence of routes that visits each city exactly once and returns to the starting city. The **Optimization** objective is to minimize the total traveling time while covering all cities. The **Prediction** task is to forecast the traveling time on each edge. This setting is more practical than previous ones used in ML4CO [48; 56], as ours considers the dynamic nature of travel costs affected by factors such as weather, road conditions, and congestion, rather than the conventional use of Euclidean distance as the cost metric between cities. We referred to the literature[57; 15] to generate raw features and traveling time on each edge. We evaluate TSP with 20 cities, adopt a 4-layer MLP with ReLU activation for prediction, and the heuristic algorithm LKH3 [27] as the solver.

Variations in road network topology [58] are common in real-world scenarios. We generate topological distributions referred from previous literature [7; 31] [2] as illustrated in Fig 2, and explore how these affect optimization on graphs, particularly on TSP. For experiments, we generate train, validation, and test topology with cluster, Gaussian, and uniform distribution, respectively. Details of data generation are specified in Appendix C.2.3.

As indicated in Table 5, compared to the IID setting, both the two-stage and SPO models exhibit significant degradation with notably larger regret in the OOD setting. However, the Inv-PnCO framework substantially mitigates this issue in the OOD setting, suggesting that learned invariant features greatly enhance generalizability against distribution shifts. Furthermore, we note that SPO performs comparably to the two-stage approach, albeit slightly better, possibly due to inherent challenges in learning invariant features for decision-making on graphs.

## 6 CONCLUSION

We propose an invariant predict-and-optimize framework, Inv-PnCO, to improve the out-of-distribution generalizability. We learn invariant decision-oriented model via a novel loss function that plugs in current PtO and PnO models and provides theoretical analysis to measure the generalization error. Experiments on various shifts on diverse combinatorial problems demonstrate the effectiveness of Inv-PnCO. We discuss limitations and broader impacts in Sec E and Appendix D .

---

[2]https://github.com/jakobbossek/tspgen

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

# A DETAILED RELATED WORK

## A.1 OPTIMIZATION UNDER UNCERTAIN COEFFICIENTS, AND PREDICT-AND-OPTIMIZE

Although neural networks have achieved notable advancements in the realm of machine learning, there remains considerable potential for enhancement in addressing optiSEization challenges. While a group of works [48; 32; 56] has been dedicated to leveraging neural networks for tackling combinatorial optimization problems under deterministic settings such as TSP, another crucial area that recently emerged is the integration of machine learning with optimization methodologies to address problems characterized by uncertain coefficients. [6] initialize the work towards combining predictive and prescriptive analysis for the optimization under uncertainty. An influential work is SPO [15] that proposes subgradient-based surrogate functions for linear optimization problems to replace non-differentiable regret functions, as well as a later extended work SPO-relax [42] for a combinatorial counterpart based on continuous relaxation, which is adopted in our experiments. We also note that the focus of these works is more on predictive models before the optimization solver, while the solvers are often treated as default heuristics (such as LKH3 [27], Dijkstra [14]) or commercial solvers (like Gurobi [23]).

In recent years, a few works on predict-and-optimize (also named decision-focused learning in the literature [62; 43; 52]) have appeared. A notable category of works utilizes the relationships of solutions of the optimization problems to learn a better predictive model. The method NCE (Noise-Contrastive Estimation) [44] designs a noise-contrastive estimation approach [24] to generate predictions, aiming for optimal solutions to achieve superior decision quality compared to non-optimal ones. The following LTR (Learning to Rank) [43] uncovers the intrinsic relationship between pairwise learning to rank in NCE, resulting in the introduction of various learn-to-rank methodologies such as pointwise rank [10], pairwise rank [33], and listwise rank [9], which aim to generate predictions that reflect the relative importance of multiple solutions.

The recent branch of work proposes learning neural network functions as surrogates for the original objective functions. Recent studies, LODL [52] and EGL [53], propose the learning of surrogate objective functions from a sample set. LANCER [69] follows a similar approach by learning surrogate functions while also incorporating optimization solving and objective function learning. SurCO [20] suggests replacing the original non-linear objective with a linear surrogate, thereby enabling the utilization of existing linear solvers.

However, some of the above methods are constrained to certain types of predict-and-optimize problems, like quadratic optimization objectives [3] or convex objectives. Though the methods based on relative importance of solutions [44; 43] and surrogate objective functions [52; 53; 69] do not constraint the type of optimizations, they require additional information such as multiple solutions or a huge number of optimization samples [52] to train the surrogate function prior to the end-to-end learning. Besides, **the most critical issue** is that most methods above are not able to run on combinatorial optimizations due to the hardness of differentiating through discrete decision variables, which makes predict-and-optimize on CO problems much harder.

In this work, in the pursuit of enhancing the generalization capabilities of predict-and-optimize in the domain of combinatorial optimization, our work endeavors to provide a general framework that is not specific to individual PnO methods. While our approach exhibits versatility across various PtO and PnO methodologies, our primary focus lies in empirically validating its efficacy under diverse problem typologies, including array-based, image-based, and graph-based scenarios, encompassing various distributional shifts. Consequently, we design experiments for our framework on one PtO model (the two-stage) and one PnO model (SPO) to facilitate the decision quality and generalization evaluation across a wide array of contexts.

## A.2 OUT-OF-DISTRIBUTION GENERALIZATION

The phenomenon of out-of-distribution generalization has garnered significant attention within the machine learning community. Pioneering works [51; 46] have explored invariant learning with causal inference. Arjovsky et al. [4] proposes invariant risk minimization (IRM) to learn an invariant representation across environments. Follow-up works propose generalizable models through the lens of group distributional robust optimization [49], game theory [2], information theory Federici

et al. [19], causal discovery [11]. The investigation of out-of-distribution (OOD) generalization has expanded across various domains, encompassing images [40], graphs [64; 63], and moleculars [67].

However, within the emerging field of machine learning for combinatorial optimization (ML4CO) uncertain coefficients, exploring the capability for out-of-distribution generalization remains largely unexplored. Particularly, though REx [37] also proposes to minimize the mean and variance terms, it is only applicable to the prediction tasks, which is validated by our experiments that learning only generalizable prediction models (i.e. the two-stage approach) is not sufficient for robust decisions, and a robust PnO model is required to utilize the information from optimizations. Our work also extends a theoretical framework for CO under uncertain coefficients, which suits both prediction-focused PtO and decision-focused PnO.

### A.3    CONNECTIONS TO RELATED DOMAINS

**Connection to (multi-source) domain adaption** In domain adaptation (DA) [59], a model is trained on labeled data (from multiple source domains) with the goal of performing well on a new, unseen target domain that has a distinct data distribution. In contrast, our scenario of out-of-distribution generalization differs from DA in the following key aspects. (1) *Information of testing distribution*: In OOD generalization, information about the target domain is unknown or unavailable. In contrast, DA assumes the target domain is known, though it may lack labels or have only a small number of labeled samples. (2) *Distribution Assumptions*: OOD generalization considers that the training and test distributions may be entirely different, while DA assumes the existence of a target domain related to the source domains, with some relationship between them.

**Connection to mutual information-based regularization in adversarial robustness** We compare with methods [72; 61; 70] that also leverages mutual information-based regularization in the field of adversarial robustness, which has difference in the research topic with ours as following: (1)*Data sample source*: Zhu et al. [72]; Wang et al. [61]; Zhou et al. [70] focus on adversarial samples, which are artificially designed through optimization algorithms to induce specific vulnerabilities in the model. Inv-PnCO addresses naturally occurring out-of-distribution (OOD) samples, which arise due to shifts between training and test data distributions. (2) *Objective*: The goal of Zhu et al. [72]; Wang et al. [61]; Zhou et al. [70] is to improve adversarial robustness by minimizing the impact of small, targeted perturbations that exploit model weaknesses, while the goal of Inv-PnCO is to enhance OOD generalization by capturing invariant factors that improve robust performance across distribution shifts. (3) *Role of mutual information*: For Zhu et al. [72]; Wang et al. [61]; Zhou et al. [70], mutual information is used to enhance the model's awareness of adversarial patterns, which mostly uses the mutual information between (adversarial) input and output, making it less susceptible to targeted perturbations. For Inv-PnCO, we employ mutual information $I_{e,q}(z; e \mid y)$ between final solution with environment $e$ given prediction $y$ to learn invariant features, which is converted to a variance term among losses of multiple training environments for a tractable loss.

In summary, the objectives and application scenarios and use of mutual information are fundamentally different where Zhu et al. [72]; Wang et al. [61]; Zhou et al. [70] focus on enhancing adversarial robustness, whereas our work centers on improving OOD generalization.

## B    PROOFS

### B.1    NOTATIONS USED IN PROOFS

Besides the definition of KL divergence given in Eq 5, we define Jensen–Shannon (JS) divergence for the raw feature-solution pair $(x, z)$ for the proofs below:

$$D_{JSD}\left(p_1(\mathbf{z}|\mathbf{x})\|p_2(\mathbf{z}|\mathbf{x})\right) = \frac{1}{2}D_{KL}\left(p_1(\mathbf{z}|\mathbf{x})\|p_m(\mathbf{z}|\mathbf{x})\right) + \frac{1}{2}D_{KL}\left(p_2(\mathbf{z}|\mathbf{x})\|p_m(\mathbf{z}|\mathbf{x})\right). \quad (10)$$

with $p_m(\mathbf{z}|\mathbf{x}) = \frac{1}{2}p_1(\mathbf{z}|\mathbf{x}) + \frac{1}{2}p_2(\mathbf{z}|\mathbf{x})$, and we abbreviate $p_e(\mathbf{z}|\mathbf{x}) = p_e(\mathbf{z}|\mathbf{x}, \mathbf{e} = e)$.

The following defines mutual information $I_e(\mathbf{z}; \mathbf{e}|\mathbf{y})$ of final decisions $\mathbf{z}$ and environment $\mathbf{e}$ conditioned on optimization coefficients $\mathbf{y}$ used in the regularization term mentioned in Theorem 1.

$$I_e(\mathbf{z}; \mathbf{e}|\mathbf{y}) = D_{KL}\left(p(\mathbf{z}|\mathbf{y}, \mathbf{e})\|p(\mathbf{z}|\mathbf{y})\right), \quad (11)$$

and the mutual information $I_{\mathbf{e},q}\left(\mathbf{x};\mathbf{z}|\mathbf{y}\right)$ mentioned in the conditions of Theorem 1 is given by:

$$I_{\mathbf{e},q}\left(\mathbf{x};\mathbf{z}|\mathbf{y}\right) = D_{KL}\left(q(\mathbf{z}|\mathbf{x},\mathbf{y})\|q(\mathbf{z}|\mathbf{y})\right). \tag{12}$$

### B.2 PROOF TO PROPOSITION 1

*Proof.* We initiate the proof by establishing the equivalence of two terms, respectively.

To begin with, for the regularization term $R(q(\mathbf{y}|\mathbf{x}))$ under the invariance condition in Assumption 1 we have:

$$
\begin{aligned}
&R(q(\mathbf{y}|\mathbf{x})) = I(\mathbf{z};\mathbf{e}|\mathbf{y})\\
=&D_{KL}(q(\mathbf{z}|\mathbf{y},\mathbf{e})\|q(\mathbf{z}|\mathbf{y}))\\
=&D_{KL}\left(q(\mathbf{z}|\mathbf{y},\mathbf{e})\|\mathbb{E}_e[q(\mathbf{z}|\mathbf{y},\mathbf{e})]\right)\\
=&\mathbb{E}_e\mathbb{E}_{z\sim p_e(\mathbf{z}|\mathbf{x}),y\sim q(\mathbf{y}|\mathbf{x})}\quad \log\frac{q(\mathbf{z}=z|\mathbf{y}=y,\mathbf{e}=e)}{\mathbb{E}_e q(\mathbf{z}=z|\mathbf{y}=y,\mathbf{e}=e)}\\
=&\mathbb{E}_e\mathbb{E}_{y\sim q(\mathbf{y}|\mathbf{x})}\left(\log\mathbb{E}_{z\sim q(\mathbf{z}|\mathbf{x})}q(\mathbf{z}=z|\mathbf{y}=y,\mathbf{e}=e) - \log\mathbb{E}_{z\sim q(\mathbf{z}|\mathbf{x})}\mathbb{E}_e q(\mathbf{z}=z|\mathbf{y}=y,\mathbf{e}=e)\right)\\
\leqslant&\mathbb{E}_e\mathbb{E}_{y\sim q(\mathbf{y}|\mathbf{x})}\left|\log\mathbb{E}_{z\sim q(\mathbf{z}|\mathbf{x})}q(\mathbf{z}=z|\mathbf{y}=y,\mathbf{e}=e) - \log\mathbb{E}_{z\sim q(\mathbf{z}|\mathbf{x})}\mathbb{E}_e q(\mathbf{z}=z|\mathbf{y}=y,\mathbf{e}=e)\right|\\
=&\mathbb{E}_e\mathbb{E}_{y\sim q(\mathbf{y}|\mathbf{x})}\left|\log\mathbb{E}_{z\sim q(\mathbf{z}|\mathbf{x})}q(\mathbf{z}=z|\mathbf{y}=y,\mathbf{e}=e) - \mathbb{E}_e\log\mathbb{E}_{z\sim q(\mathbf{z}|\mathbf{x})}q(\mathbf{z}=z|\mathbf{y}=y,\mathbf{e}=e)\right|\\
\leqslant&\sqrt{\mathbb{E}_e\left[|\mathbb{E}_{y\sim q(\mathbf{y}|\mathbf{x})}\log\mathbb{E}_{z\sim q(\mathbf{z}|\mathbf{x})}q(\mathbf{z}=z|\mathbf{y}=y,\mathbf{e}=e) - \mathbb{E}_e\mathbb{E}_{y\sim q(\mathbf{y}|\mathbf{x})}\log\mathbb{E}_{z\sim q(\mathbf{z}|\mathbf{x})}q(\mathbf{z}=z|\mathbf{y}=y,\mathbf{e}=e)|^2\right]}\\
=&\sqrt{\mathbb{E}_e\left[|\mathcal{L}(\mathbf{x},\mathbf{y},\mathbf{z}) - \mathbb{E}_e\mathcal{L}(\mathbf{x},\mathbf{y},\mathbf{z})|^2\right]}\\
=&\sqrt{\text{Var}_e[\mathcal{L}(\mathbf{x},\mathbf{y},\mathbf{z})]}
\end{aligned}
\tag{13}
$$

where the third step is given by:

$$
\begin{aligned}
&D_{KL}\left(p(\mathbf{z}|\mathbf{y})\|\mathbb{E}_e q(\mathbf{z}|\mathbf{y})\right) - D_{KL}(q(\mathbf{z}|\mathbf{y})\|p(\mathbf{z}|\mathbf{y},\mathbf{e})) - D_{KL}\left(\mathbb{E}_e p\left(\mathbf{z}|\mathbf{y},\mathbf{e}\right)\|\mathbb{E}_e[q(\mathbf{z}|\mathbf{y})]\right)\\
=&\mathbb{E}_{q(\mathbf{z}|\mathbf{y})}\log\frac{q(\mathbf{z}|\mathbf{y})}{\mathbb{E}_e q(\mathbf{z}|\mathbf{y})} - \mathbb{E}_{q(\mathbf{z}|\mathbf{y})}\log\frac{q(\mathbf{z}|\mathbf{y})}{p(\mathbf{z}|\mathbf{y},\mathbf{e})} - \mathbb{E}_{\mathbb{E}_e p(\mathbf{z}|\mathbf{y},\mathbf{e})}\log\frac{\mathbb{E}_e p(\mathbf{z}|\mathbf{y},\mathbf{e})}{\mathbb{E}_e q(\mathbf{z}|\mathbf{y})}\\
=&\mathbb{E}_{q(\mathbf{z}|\mathbf{y})}\log\frac{p(\mathbf{z}|\mathbf{y},e)}{\mathbb{E}_e q(\mathbf{y}|\mathbf{z})} - \mathbb{E}_e p(\mathbf{y}|\mathbf{z},\mathbf{e})\log\frac{\mathbb{E}_e p(\mathbf{y}|\mathbf{z},\mathbf{e})}{\mathbb{E}_e q(\mathbf{y}|\mathbf{z})}\\
=&\mathbb{E}_p\log\frac{p(\mathbf{z}|\mathbf{y},\mathbf{e})}{\mathbb{E}_e p(\mathbf{z}|\mathbf{y},\mathbf{e})}\\
=&D_{KL}\left(p(\mathbf{z}|\mathbf{y},\mathbf{e})\|\mathbb{E}_e[p(\mathbf{z}|\mathbf{y},\mathbf{e})]\right)
\end{aligned}
\tag{14}
$$

The last inequality is due to the Cauchy-Schwarz inequality, and the equality holds when $q(z|y)$ is delta distribution (i.e., deterministic solver).

Then for the $D_{KL}(p(\mathbf{z}|\mathbf{x},\mathbf{e})\|q(\mathbf{z}|\mathbf{y}))$ term we have:

$$
\begin{aligned}
&D_{KL}(p(\mathbf{z}|\mathbf{x},\mathbf{e})\|q(\mathbf{z}|\mathbf{y}))\\
=&\mathbb{E}_e\mathbb{E}_{z\sim p_e(\mathbf{z}|\mathbf{x}=x),y\sim q(\mathbf{y}|\mathbf{x}=x),x\sim p_e(\mathbf{x})}\log\frac{p(\mathbf{z}=z|\mathbf{x}=x,\mathbf{e}=e)}{q(\mathbf{z}=z|\mathbf{y}=y)}\\
\leqslant&4
\end{aligned}
\tag{15}
$$

where $\mathcal{L}_e(x,y,z)$ is the decision oriented loss for the data generated by the environment $e$, the last inequality is given by Jensen's Inequality, and the equality holds when $q(\mathbf{z}|\mathbf{y})$ is a delta distribution (deterministic solver). Then $\min_{\mathcal{M}}\mathbb{E}_e[\mathcal{L}(\mathbf{x},\mathbf{y},\mathbf{z})]$ is the upper bound of $\min_{\mathcal{M}}D_{KL}\left(p_e(\mathbf{z}|\mathbf{x})\|q(\mathbf{z}|\mathbf{y})\right)$. Since we also have $\min_{\mathcal{M}}\text{Var}_e[\mathcal{L}(\mathbf{x},\mathbf{y},\mathbf{z})]$ is the upper-bound for $\min_{\mathcal{M}}I(\mathbf{z};\mathbf{e}|\mathbf{y})$ by the above, this completes the proof. $\square$

### B.3 PROOF TO THEOREM 1

Before the full proof to Theorem 1, we give the following lemmas extended the results to distributions between raw features $\mathbf{x}$ and final solutions $\mathbf{z}$ from the propositions in Federici et al. [19].

**Lemma 1.** *For any predictor $q(\mathbf{y}|\mathbf{x})$ and solver $q(\mathbf{z}|\mathbf{y})$ during training environment factor $e$ and testing environment factor $e'$, we have*

$$
\begin{aligned}
D_{KL}\left(p_e(\mathbf{z}|\mathbf{x})\|q(\mathbf{z}|\mathbf{x})\right) &\leqslant I_e(\mathbf{x};\mathbf{z}|\mathbf{y}) + D_{KL}\left(p_e(\mathbf{z}|\mathbf{y})\|q(\mathbf{z}|\mathbf{y})\right) \\
D_{KL}\left(p_{e'}(\mathbf{z}|\mathbf{x})\|q(\mathbf{z}|\mathbf{x})\right) &\leqslant I_{e'}\left(\mathbf{x};\mathbf{z}|\mathbf{y}\right) + D_{KL}\left(p_{e'}(\mathbf{z}|\mathbf{y})\|q(\mathbf{z}|\mathbf{y})\right)
\end{aligned}
\tag{16}
$$

*Proof.* During the training stage with the environment factor $e$, we have:

$$
\begin{aligned}
&D_{KL}\left(p_e(\mathbf{z}|\mathbf{x})\|q(\mathbf{z}|\mathbf{x})\right) \\
=&\mathbb{E}_{x\sim p_e(\mathbf{x})}\left[\mathbb{E}_{z\sim p_e(\mathbf{z}|\mathbf{x}=x)}\log\frac{p_e(\mathbf{z}=z|\mathbf{x}=x)}{q(\mathbf{z}=z|\mathbf{x}=x)}\right] \\
=&\mathbb{E}_{x\sim p_e(\mathbf{x})}\left[\mathbb{E}_{z\sim p_e(\mathbf{z}|\mathbf{x}=x)}\log\frac{p_e(\mathbf{z}=z|\mathbf{x}=x)}{\mathbb{E}_{y\sim q(\mathbf{y}|\mathbf{x}=x)}q(\mathbf{z}=z|\mathbf{y}=y)}\right] \\
\leqslant&\mathbb{E}_{x\sim p_e(\mathbf{x})}\left[\mathbb{E}_{z\sim p(\mathbf{z}|\mathbf{x}=x)}\mathbb{E}_{y\sim q(\mathbf{y}|\mathbf{x}=x)}\log\frac{p_e(\mathbf{z}=z|\mathbf{x}=x)}{q(\mathbf{z}=z|\mathbf{y}=y)}\right] \\
=&D_{KL}\left(p_e(\mathbf{z}|\mathbf{x})\|q(\mathbf{z}|\mathbf{y})\right)
\end{aligned}
\tag{17}
$$

where the third step is according to Jensen's Inequality and the equality holds when $q(\mathbf{y}|\mathbf{x})$ is a delta distribution (deterministic predictor). The above term could continue as:

$$
\begin{aligned}
&D_{KL}\left(p_e(\mathbf{z}|\mathbf{x})\|q(\mathbf{z}|\mathbf{y})\right) \\
=&\mathbb{E}_{x\sim p_e(\mathbf{x})}\left[\mathbb{E}_{z\sim p(\mathbf{z}|\mathbf{x}=x)}\mathbb{E}_{y\sim q(\mathbf{y}|\mathbf{x}=x)}\log\frac{p_e(\mathbf{z}=z|\mathbf{x}=x)}{p_e(\mathbf{z}=z|\mathbf{y}=y)}\cdot\frac{p_e(\mathbf{z}=z|\mathbf{y}=y)}{q(\mathbf{z}=z|\mathbf{y}=y)}\right] \\
=&\mathbb{E}_{x\sim p_e(\mathbf{x}),z\sim p(\mathbf{z}|\mathbf{x}=x),y\sim q(\mathbf{y}|\mathbf{x}=x)}\log\frac{p(\mathbf{x},\mathbf{z}|\mathbf{y})}{p(\mathbf{x}|\mathbf{y})p(\mathbf{z}|\mathbf{y})} + \mathbb{E}_{p_e(\mathbf{z}|\mathbf{y})}\log\frac{p_e(\mathbf{z}|\mathbf{y})}{q(\mathbf{z}|\mathbf{y})} \\
=&I(\mathbf{z};\mathbf{x}|\mathbf{y}) + D_{KL}\left(p_e(\mathbf{z}|\mathbf{y})\|q(\mathbf{z}|\mathbf{y})\right)
\end{aligned}
\tag{18}
$$

The inequality with the test environment factor $e'$ holds similarly to the above case of the training environment factor $e$, which completes the proof.

$\square$

The following lemma gives JS-divergence of induced solver $q(\mathbf{z}|\mathbf{y})$ and distribution of $p_{e'}(\mathbf{z}|\mathbf{y})$ under environment $e'$.

**Lemma 2.**

$$
D_{JSD}\left(p_{e'}(\mathbf{z}|\mathbf{y})\|q(\mathbf{z}|\mathbf{y})\right) \leq \left(\sqrt{\frac{1}{2\alpha}I(\mathbf{z};\mathbf{e}|\mathbf{y})} + \sqrt{\frac{1}{2}D_{KL}\left(p_e(|\mathbf{y})\right)\|q(\mathbf{z}|\mathbf{y})}\right)^2
\tag{19}
$$

*Proof.* To begin with, we have:

$$
\begin{aligned}
&I(\mathbf{z};\mathbf{e}|\mathbf{y}) \\
=&\ D_{KL}(p(\mathbf{z}|\mathbf{y},\mathbf{e})\|p(\mathbf{z}|\mathbf{y})) \\
\geqslant&\ 2\alpha\left(\frac{1}{2}D_{KL}\left(p_e(\mathbf{z}|\mathbf{y})\|q(\mathbf{z}|\mathbf{y})\right) + \frac{1}{2}D_{KL}\left(p_e(\mathbf{z}|\mathbf{y})\|q(\mathbf{z}|\mathbf{y})\right)\right) \\
=&\ 2\alpha D_{JSD}\left(p_e(\mathbf{z}|\mathbf{y})\|p_e(\mathbf{z}|\mathbf{y})\right) + D_{KL}\left(p_m(\mathbf{z}|\mathbf{y})\|p(\mathbf{z}|\mathbf{y})\right) \\
\geqslant&\ 2\alpha D_{JSD}\left(p_e(\mathbf{z}|\mathbf{y})\|p_{e'}(\mathbf{z}|\mathbf{y})\right)
\end{aligned}
\tag{20}
$$

where $p_m(\mathbf{z}|\mathbf{y}) = \frac{1}{2}p_e(\mathbf{z}|\mathbf{y}) + \frac{1}{2}p_{e'}(\mathbf{z}|\mathbf{y})$.

Besides, as the square root of the Jensen-Shannon divergence is a metric [17], by triangle inequality:

$$
\sqrt{D_{JSD}\left(p_e(\mathbf{z}|\mathbf{y})\|q(\mathbf{z}|\mathbf{y})\right)} + \sqrt{D_{JSD}\left(p_{e'}(\mathbf{z}|\mathbf{y})\|p_e(\mathbf{z}|\mathbf{y})\right)} \geqslant \sqrt{D_{JSD}\left(p_{e'}(\mathbf{z}|\mathbf{y})\right)\|q(\mathbf{z}|\mathbf{y})}
\tag{21}
$$

In addition, we are able to bound the JS-divergence in terms of KL-divergence as:

$$D_{JSD}\left(p_e(\mathbf{z}|\mathbf{y})\|q(\mathbf{z}|\mathbf{y})\right) = \frac{1}{2}D_{\mathrm{KL}}\left(p_e(\mathbf{z}|\mathbf{y})\|q(\mathbf{z}|\mathbf{y})\right) - D_{\mathrm{KL}}\left(p_m(\mathbf{z}|\mathbf{y})\|q(\mathbf{z}|\mathbf{y})\right)$$
$$\leqslant \frac{1}{2}D_{KL}\left(p_e(\mathbf{z}|\mathbf{y})\|q(\mathbf{z}|\mathbf{y})\right) \tag{22}$$

In conclusion, with the above three inequalities, we have:

$$D_{JSD}\left(p_{e'}(\mathbf{z}|\mathbf{y})\|q(\mathbf{z}|\mathbf{y})\right)$$
$$\leqslant \left(\sqrt{D_{JSD}\left(p_e(\mathbf{z}|\mathbf{y})\|q(\mathbf{z}|\mathbf{y})\right)} + \sqrt{D_{JSD}\left(p_{e'}(\mathbf{z}|\mathbf{y})\|p_e(\mathbf{z}|\mathbf{y})\right)}\right)^2$$
$$\leqslant \left(\sqrt{\frac{1}{2}D_{KL}\left(p_e(\mathbf{z}|\mathbf{y})\|q(\mathbf{z}|\mathbf{y})\right)} + \sqrt{\frac{1}{2\alpha}I(\mathbf{z};\mathbf{e}|\mathbf{y})}\right)^2 \tag{23}$$

where the second line is according to Eq (21), and the first and second term in the third line is according to Eq (22) and Eq (20), respectively. □

**Lemma 3.**

$$\min_{q(\mathbf{z}|\mathbf{y})} D_{KL}\left(p_e(\mathbf{z}|\mathbf{x})\|q(\mathbf{z}|\mathbf{y})\right) \Leftrightarrow \min_{q(\mathbf{y}|\mathbf{x}),q(\mathbf{z}|\mathbf{y})} I_e(\mathbf{x};\mathbf{z}|\mathbf{y}) + D_{KL}\left(p_e(\mathbf{z}|\mathbf{y})\|q(\mathbf{z}|\mathbf{y})\right) \tag{24}$$

*Proof.* Regarding the mutual information term $I(\mathbf{x};\mathbf{z}|\mathbf{y})$, we have:

$$\min_{q(\mathbf{y}|\mathbf{x}),q(\mathbf{z}|\mathbf{y})} I(\mathbf{x};\mathbf{z}|\mathbf{y}) = \min_{q(\mathbf{y}|\mathbf{x}),q(\mathbf{z}|\mathbf{y})} \mathbb{E}_{x,y,z} \log \frac{p(\mathbf{x},\mathbf{z}|\mathbf{y})}{p(\mathbf{x}|\mathbf{y})q(\mathbf{z}|\mathbf{y})}$$
$$= \min_{q(\mathbf{y}|\mathbf{x}),q(\mathbf{z}|\mathbf{y})} \mathbb{E}_{x,y,z} \log \frac{p(\mathbf{z}|\mathbf{x},\mathbf{y})}{q(\mathbf{z}|\mathbf{y})} \tag{25}$$
$$= \min_{q(\mathbf{y}|\mathbf{x}),q(\mathbf{z}|\mathbf{y})} D_{KL}\left(p_e(\mathbf{z}|\mathbf{x})\|q(\mathbf{z}|\mathbf{y})\right) - D_{KL}(p(\mathbf{z}|\mathbf{y})\|q(\mathbf{z}|\mathbf{y}))$$

which is equivalent to that in the lemma and completes the proof. □

Based on the above lemmas, we are able to arrive at the proof for Theorem 1.

*Proof.* According to Proposition 1, minimizing the loss function in Eq 8 is equivalent for minimizing:

$$I(\mathbf{y};\mathbf{e}|\mathbf{z}) + D_{KL}\left(p_e(\mathbf{z}|\mathbf{x})\|q(\mathbf{z}|\mathbf{y})\right), \tag{26}$$

and according to Lemma 3, is further equivalent to:

$$\min \underbrace{I(\mathbf{y};\mathbf{e}|\mathbf{z})}_{①} + \underbrace{D_{KL}\left(p_e(\mathbf{z}|\mathbf{y})\|q(\mathbf{z}|\mathbf{y})\right)}_{②} + \underbrace{I_e(\mathbf{x};\mathbf{z}|\mathbf{y})}_{③} \tag{27}$$

According to Lemma 2, minimizing $D_{JSD}\left(p_{e'}(\mathbf{z}|\mathbf{y})\|q(\mathbf{z}|\mathbf{y})\right)$ is equivalent to minimize the lower bound for ① and ②. Additionally for ③, we have the following equation:

$$D_{KL}\left(p_e(\mathbf{z}|\mathbf{y})\|p_e(\mathbf{z}|\mathbf{y},\mathbf{x})\right) = D_{KL}\left(p_{e'}(\mathbf{z}|\mathbf{y})\|p_{e'}(\mathbf{z}|\mathbf{y},\mathbf{x})\right) \tag{28}$$

could be satisfied when minimizing $D_{KL}(p(\mathbf{z}|\mathbf{x})\|q(\mathbf{z}|\mathbf{y}))$, then we can reach

$$I_e(\mathbf{x};\mathbf{z}|\mathbf{y}) = I_{e'}(\mathbf{x};\mathbf{z}|\mathbf{y}). \tag{29}$$

By combining Lemma 1 and Eq (29), minimizing ①, ② and ③ is equivalent to minimizing $I_{e'}(\mathbf{x};\mathbf{z}|\mathbf{y}) + D_{KL}\left(p_{e'}(\mathbf{z}|\mathbf{y})\|q(\mathbf{z}|\mathbf{y})\right)$, i.e. $D_{KL}\left(p_{e'}(\mathbf{z}|\mathbf{x})\|q(\mathbf{z}|\mathbf{x})\right)$, which completes the proof. □

## C  EXPERIMENT DETAILS

We specify experimental details in this section. Code will be released after publication.

## C.1 MODEL DETAILS

### C.1.1 PREDICTION MODELS

**Multi-layer perceptron (MLP)** To ensure a fair comparison, within the same task, we adopt the same prediction model for predicting optimization coefficient. We adopt multi-layer perceptron (MLP) for the knapsack and TSP tasks. The predictive model $\mathcal{M}$ using MLP is formulated as follows:

$$\mathbf{a}^{(i+1)} = \sigma(\mathbf{W}^{(i)}\mathbf{a}^{(i)} + \mathbf{b}^{(i)}), \quad i = 1, 2, \ldots, K-1, \tag{30}$$

where $\mathbf{a}^{(1)} = \mathbf{x}$ and $\mathbf{y} = \mathbf{a}^{(K)}$ represent the input and output for $\mathcal{M}$ respectively. Here, $a^i$ denotes the hidden vector for $i = 2, \cdots, K-1$, $b$ signifies the bias term, and $\sigma$ denotes the activation function, specifically ReLU in our case. In our experiments, we set the size of intermediate hidden units to 32 and utilize $K = 3$ layers in the knapsack problem and $K = 4$ for the TSP task.

**Resnet-18** We adopt the ResNet [26] in the torchvision [39] package for the prediction of the visual shortest path task. ResNet-18 serves as a popular baseline model in many research studies and benchmark datasets, making it an essential component of contemporary deep learning research in computer vision.

### C.1.2 DECISION MODELS

In the training stage, we train the model by the the respective loss of PtO (two-stage) or PnO (SPO) approach specified below. In the testing stage, we first predict the coefficients using the predictive model, then adopt the respective solver for the forward pass to obtain the decisions using the predicted coefficients, and evaluate the decision quality with regret.

**The two-stage approach** The two-stage approach, as specified as a model in "predict-then-optimize" that is trained towards the goal of "prediction optimal" (in def 1), directly trains the loss to optimize the prediction of optimization coefficients. As all involved predictions are regression tasks, the loss function is specified as Mean Squared Error (MSE) to quantify the dissimilarity between predicted ($\hat{y}$) and actual ($y$).

$$\text{MSE}(\hat{\mathbf{y}}, \mathbf{y}) = \frac{1}{n} \sum_{i=1}^{n} (\mathbf{y}_i - \hat{\mathbf{y}}_i)^2 \tag{31}$$

MSE is defined as the average squared difference between predicted and actual values across a dataset of size $n$.

**The SPO method** The SPO, as specified as a model in "predict-and-optimize" that is trained towards the goal of "decision optimal" (in def 2), trains a subgradient-based surrogate function of the regret function to optimize the decision quality instead of the prediction task. We train the model using Eq. 32 as the loss function, and in the backward pass, the prediction model is updated by its continuous relaxation. Specifically, the surrogate loss function for **SPO** [42] that is used in our experiments is:

$$\mathcal{L}_{spo}(\mathbf{y}, \mathbf{z}, \hat{\mathbf{y}}, \hat{\mathbf{z}}) = -\mathcal{F}(\tilde{\hat{\mathbf{z}}}, 2\hat{\mathbf{y}} - \mathbf{y}) + 2\mathcal{F}(\mathbf{z}, \hat{\mathbf{y}}) - \mathcal{F}(\mathbf{z}, \mathbf{y}) . \tag{32}$$

where $\mathbf{z}$ denotes the optimal solution using the ground-truth coefficient $\mathbf{y}$, and $\tilde{\hat{\mathbf{z}}}$ denotes solution obtained with the coefficient $(2\hat{\mathbf{y}} - \mathbf{y})$.

## C.2 DETAILED DATASETS AND ENVIRONMENT ACQUISITION

We list the distributions used for each dataset in Table 6, as well as the acquired environments.

### C.2.1 KNAPSACK PROBLEM WITH UNKNOWN PROFITS

We adopt the problem from the previous literature [13; 42; 41; 44; 22]. The raw features $x$ and profits $y$ in Knapsack dataset $(x_1, y_1), (x_2, y_2), \ldots, (x_n, y_n)$ is generated according to the polynomial function as described in prior literature [15]:

$$y_i = \left[ \frac{1}{3.5^{\deg}\sqrt{p}} \left( (\mathcal{B}x_i) + 3 \right)^{\deg} + 1 \right] \cdot \epsilon_i, \tag{33}$$

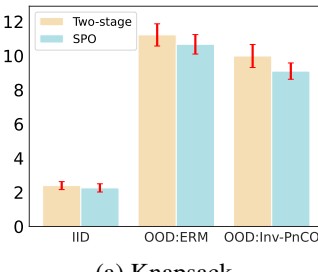 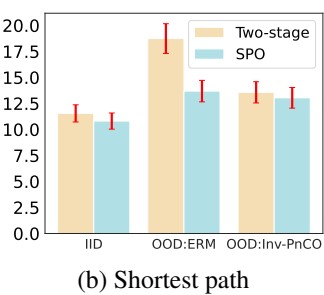 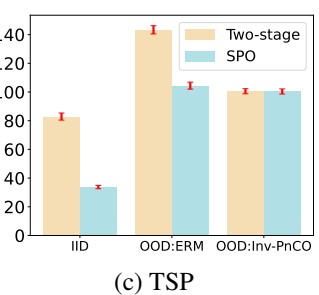

|(a) Knapsack | (b) Shortest path | (c) TSP |

Figure 4: Result of decision quality in regret with error bars of each optimization task. Each figure specifies the results under i.i.d. setting (denoted "IID"), and OOD setting of ERM and Inv-PnCO.

Table 6: Distributions for training, validation, testing, and acquired environments of Inv-PnCO. Knapsack problems adopt Gaussian distribution; the parameters specify mean and standard deviation (std). The visual shortest path adopts image augmentations upon the original graphs, where the parameters specify the type of augmentations and corresponding value. TSP adopts different graph typologies and is elaborated in Appendix C.2.3. The last line specifies the hyper-parameters for the best result of Inv-PnCO shown as (number of environments, $\beta$, learning rate). SP problem specifies $L_1$ regularization weight additionally.

| | Knapsack | Shortest path | TSP |
|---|---|---|---|
| Predictor | MLP | Resnet-18 [26] | MLP |
| Solver | Gurobi [23] | Dijkstra [14] | LKH3 [27] |
| Shift | covariate shift | concept shift | covariate shift |
| Train | gaussian $(10, 10)$ | contrast $(10)$ | cluster $(4, (15, 55) \pm 15)$ |
| Validation | gaussian $(5, 5)$ | hue $(0.3)$ | gaussian $(50, 10 \pm 5)$ |
| Test | gaussian $(0, 1)$ | Original | uniform $(30, 40)$ |
| Acquired Environments in Inv-PnCO | env0: gaussian $(32, 1)$
env1: gaussian $(16, 1)$
env2: gaussian $(8, 1)$
env3: gaussian $(4, 1)$
env4: gaussian $(2, 1)$ | env0: saturation $(1)$
env1: brightness $(1)$
env2: contrast $(3)$
env3: brightness $(3)$
env4: contrast $(5)$ | env0: explosion
  $((20, 60), (37, 43), (5, 7))$
env1: cluster $(2, (20, 40) \pm 7)$
env2: gaussian $(40, 60)$ |
| Best hyper parameters | MSE: (5, 4.0, 1e-2)
SPO: (1, 1.0, 5e-2) | MSE: (2, 4.0, 1e-5, 1e-5)
SPO: (2, 0.5, 1e-4, 0) | MSE: (3, 4.0, 1e-3)
SPO: (3, 0.5, 5e-3) |

where each $x_i \sim N(\mu, \sigma * I_p)$ is drawn from a multivariate Gaussian distribution, (where $\mu$ and $\sigma$ are parameters controlling the distribution) the matrix $\mathcal{B}^* \in \mathbb{R}^{d \times p}$ encodes the parameters of the true model, with each entry of $\mathcal{B}^\star$ being a Bernoulli random variable that equals 1 with a probability of 0.5. $\epsilon_i^j$ represents a multiplicative noise term with a uniform distribution, and $p$ denotes the given number of features. The weights of the knapsack problem are fixed and sampled uniformly from the range of 3 to 8. For our experiments, we set the default capacity to 30, and the number of items to 20. We utilize a polynomial degree $\deg$ of 4, the dimension of raw feature is set as 5, and the random noise $\epsilon_i^j$ is sampled within the uniform distribution $\mathcal{U}(1 - w, 1 + w)$ with as $w = 0.5$. The seed is set as 2023.

For the distributions among different sets, as shown in Table 6, the training dataset adopts the Gaussian distribution $\mathcal{N}(10, 10)$ with a mean of 10 and standard deviation (std) of 10, while the distribution of validation and testing sets are $\mathcal{N}(5, 5)$ and $\mathcal{N}(0, 1)$.

### C.2.2 VISUAL SHORTEST PATH (SP) PLANNING WITH UNKNOWN COST

The visual shortest path planning task uses the publicly available Warcraft terrain map dataset [25], and we conform to the MIT License specified in the GitHub link [1]. The maps feature a grid measuring k by k, with each vertex denoting a terrain characterized by an undisclosed fixed cost to the network. A label is generated by encoding the shortest path, representing the minimum cost, from the top-left

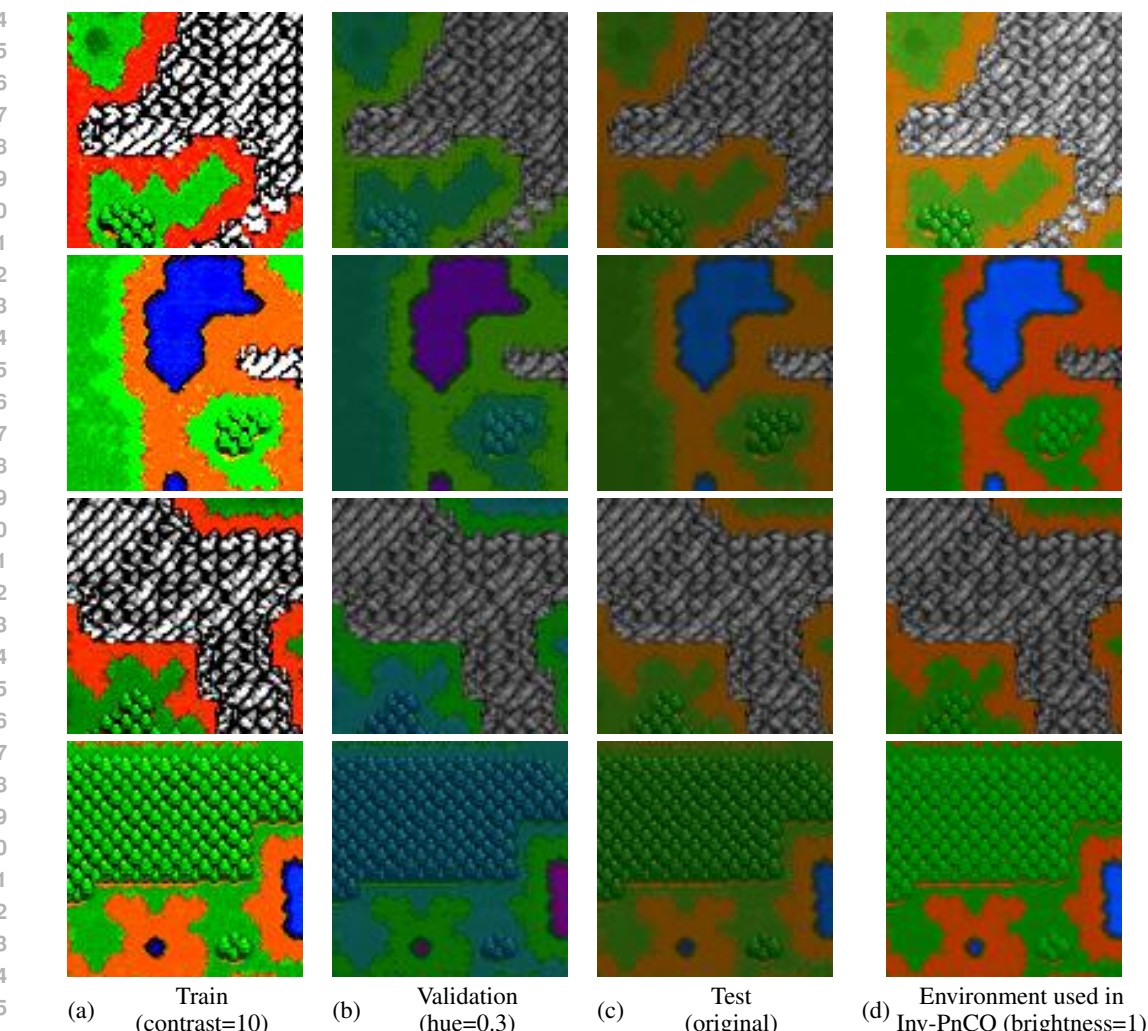

| (a) | Train (contrast=10) | (b) | Validation (hue=0.3) | (c) | Test (original) | (d) | Environment used in Inv-PnCO (brightness=1) |

Figure 5: Visualization of perceptual distribution shifts in visual shortest path problem and example of generated environments in Inv-PnCO.

to the bottom-right vertices in the form of an indicator matrix. We conduct the experiments of the shortest path problem on the $12 \times 12$ grid. The seed is set as 2023.

The distributions of each data set are included in Table 6, where we remained test set images unchanged as the original data, while the training images are augmented by "contrast" with the value of 10, and the validation images are augmented with "hue" of the value 0.3. The acquired environments are augmented in similar ways. All image augmentations are conducted by torchvision [39] package. In Fig 5, we visualize the distribution used in training, validation, and testing, as well as an example environment in Inv-PnCO. In this experiment, we employed image augmentations on the raw images while maintaining the final cost unchanged. Such a construction engenders a conceptual shift between the original data $\mathbf{x}$ and the decision $\mathbf{z}$.

### C.2.3 TRAVELLING SALESMAN PROBLEM (TSP) WITH UNKNOWN COSTS

The Traveling Salesman Problem (TSP) is a classical combinatorial optimization problem that seeks to determine the shortest possible route that visits a set of given cities exactly once and returns to the origin city. Mathematically, it can be formulated as finding the Hamiltonian cycle of minimum total length in a complete graph, where each vertex represents a city and each edge represents a path

between two cities with an associated cost or distance. This problem is renowned for its computational complexity and has numerous real-world applications in logistics, transportation, and network design.

In the domain of Machine Learning for Combinatorial Optimization (ML4CO), the recent works [48; 56] of neural solvers for the TSP often treat the Euclidean distance between cities as the direct measure of traversal time. However, in more realistic scenarios, traversal time may be contingent upon multiple factors and subject to variation with changes in features. In this study, we delve into the TSP under the unknown traversal times. While existing literature [57] has discussed TSP under uncertain coefficients, we contend that its formulation may lack coherence with real scenarios. In the modeling of [57], traversal times along edges are solely dependent upon edge-specific features, disregarding any correlation with city coordinates (i.e., Euclidean distance). Therefore, with insights from these previous studies, we propose a new simulation for modeling traversal times.

In this work, we treat the TSP as an undirected complete graph, where each city is treated as a node and each two nodes are connected. The generation of graph typologies follows previous literature [2] [35; 7] that is also adopted in the works of ML4CO [7; 31]. We initialize the node coordinates following distribution, which is specified below. The node coordinate is treated as node feature $x_u$ for node $u$, and edge feature $x_e$ includes potential factors that influence traveling time on the edge, including the road conditions (such as width, smoothness, presence of buildings with concentrated pedestrian traffic, etc.) are abstracted into a feature vector $x_e$. In our implementation, this vector $x_e$ is generated through sampling from the Gaussian distribution $\mathcal{N}(0,1)$ with a mean of 0 and a standard deviation of 1.

Then, for an edge $e = (u, v)$ with two connecting nodes $u$ and $v$, we give $d_e$ as the Euclidean distance by following:

$$d_e = D_E(x_u, x_v) \tag{34}$$

where $D_E$ denotes pairwise Euclidean distance, and the traveling time $t_e$ (cost) on each edge is constructed by:

$$t_e = d_e * c_e + \text{poly}(x_e) \tag{35}$$

where $c_e$ is the parameter on each edge indicating the road congestion, which is sampled from Gaussian distribution $\mathcal{N}(1,1)$ and takes the absolute value to be positive, and poly is the polynomial function following [15] as:

$$poly(x_e) = \frac{1}{3^{\text{deg-1}}\sqrt{p}} \left( (\mathcal{B}\boldsymbol{x}_e)_j + 3 \right)^{\text{deg}} \cdot \epsilon_j \tag{36}$$

where $\epsilon_j$ is the noise term which is sampled within the uniform distribution $\mathcal{U}(1-w, 1+w)$ and $w$ is the noise width specified as 0.2 in our experiments. The degree is set as 2 and the seed is set as 2023.

In our experiments, as shown in Fig 2, we adopt the following topological distributions to evaluate the predict-and-optimize for TSP under distribution shifts:

- **Cluster distribution** for the training set, as shown in Fig 2(a), and the distribution parameters "cluster $(4, (15, 55) \pm 15)$" means the training set is generated by nodes of 4 clusters with the centers of clusters is sampled from a uniform distribution of $\mathcal{U}(15, 55)$ where the nodes are sampled around the centers with standard deviation of 3.

- **Gaussian distribution** for the validation set, as shown in Fig 2(b), and the parameters "gaussian $(50, 10 \pm 5)$" means the nodes are generated by the Gaussian distribution where the coordinates of $x$ are sampled from $\mathcal{N}(50, 5)$ and coordinates of $y$ are sampled from $\mathcal{N}(10, 5)$.

- **Uniform distribution** for the testing set, as shown in Fig 2(c), and the parameters "uniform $(30, 40)$" means the coordinates of $x$ and $y$ are generated by the uniform distribution $\mathcal{U}(30, 40)$.

- **Explosion distribution** for the generated environment, as shown in Fig 2(d), where the parameters of "explosion $((20, 60), (37, 43), (5, 7))$" means the node coordinates are firstly generated by uniform distribution $\mathcal{U}(20, 60)$, and then generate one center of "explosion" by sampling from $\mathcal{U}(37, 43)$, where the explosion radius is sampled from $\mathcal{U}(5, 7)$ and the nodes within the radius are pushed to the borders.

We generate the node coordinates of all these distributions by the public implementation [2].

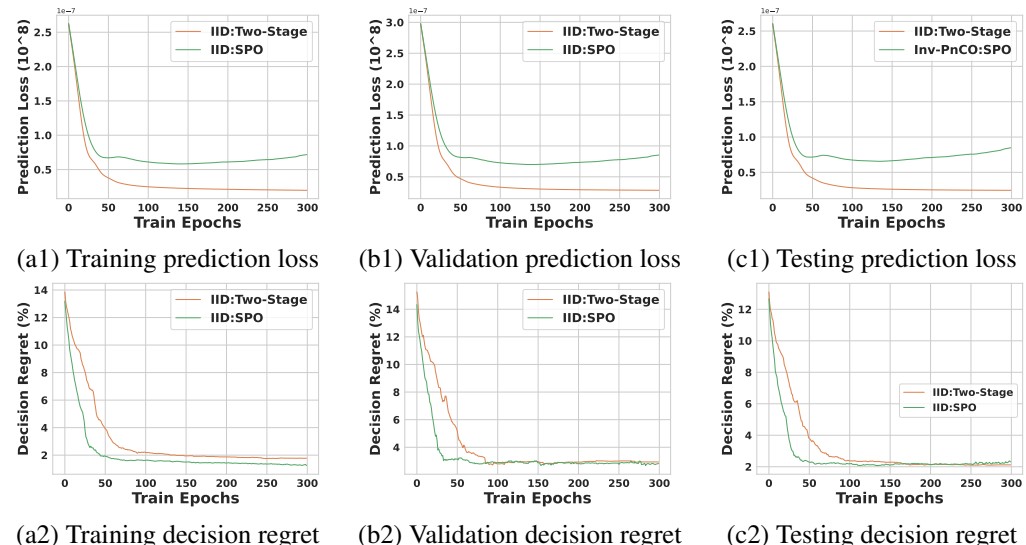

Figure 6: Prediction loss and decision quality (in regret) throughout the training of ERM in IID setting on the knapsack problem.

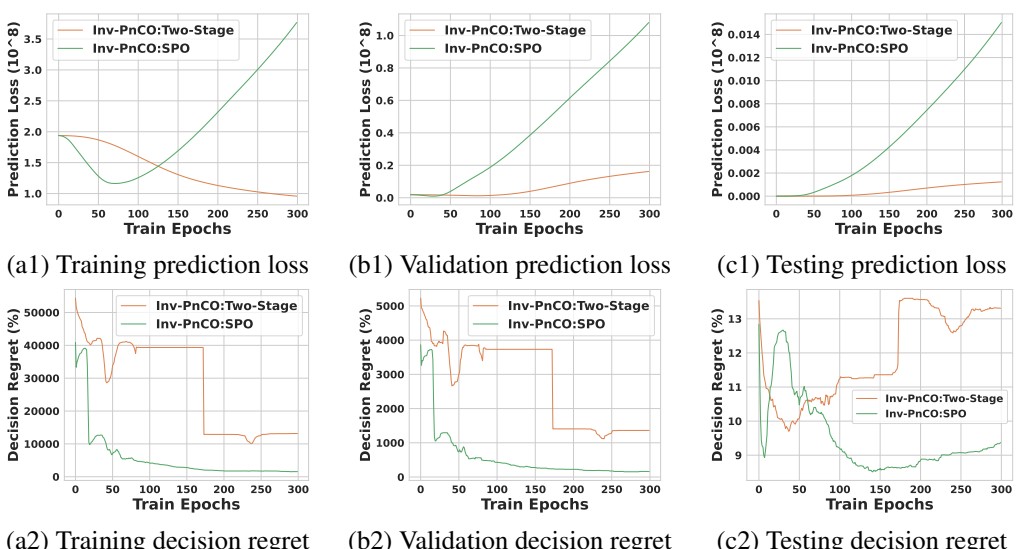

Figure 7: Prediction loss and decision quality (in regret) throughout the training of our proposed Inv-PnCO framework on the knapsack problem.

## C.3   EXPERIMENTAL RESULT DETAILS

## C.4   SENSITIVITY ANALYSIS, ABLATION STUDY, QUALITATIVE ANALYSIS AND VISUALIZATION

We present a summary of results below, where detailed **sensitivity analyses**, **Qualitative analysis visualizations**, and the **ablation study** are provided in Appendix C.3. We assess the sensitivity of Inv-PnCO framework on the knapsack problem across various optimization parameters, encompassing the constraint, the size of decision variables and training hyperparameters. Results show that Inv-PnCO framework consistently reduces regret in comparison to ERM across diverse optimization parameters.

Next, we visualize the training progression to analyze Inv-PnCO under diverse environments. As depicted in Fig 3(a), the losses of SPO for each environment decrease over training epochs. Furthermore, the losses in different environments tend to become similar, which may indicate that Inv-PnCO improves generalizability by reducing disparities of decision qualities of multiple environments. Notably, as illustrated in Figures 3(b) and (c), although SPO exhibits higher prediction loss, it yields lower regret due to its capability to learn the prediction model using the information of final objectives. This phenomenon is similar to the relationship observed between the prediction loss curve and decision quality curve in the i.i.d setting in Fig. 6 in the appendix. This also validates the necessity of designing the decision-oriented invariant learning framework Inv-PnCO compared to the generalization models of pure ML tasks.

### C.4.1 EXPERIMENT VISUALIZATIONS

We show the result of decision quality in regret with error bars of each optimization task in Fig. 4. Each figure visualizes the regret under IID and OOD (of ERM and Inv-PnCO). We note that the test sets are identical for IID and OOD settings. We observe a decline in decision quality under OOD settings and find that the proposed Inv-PnCO framework significantly reduces regret. In the TSP task, our Inv-PnCO approach also improved decision quality compared to ERM. Our Inv-PnCO's performance is comparable for SPO and the two-stage approach, this may indicate that robust decision-focused learning is more challenging for complex decision problems.

We visualize the curves of ERM under the IID setting with prediction loss and regret curves in Fig.6 during the training, validation, and testing sets. We observe that the change in regret sometimes exhibits a stepwise pattern, which could be due to the combinatorial nature of CO problems. We also note that for visualization purposes, we disabled early stopping, which resulted in SPO overfitting in the final stages. This leads to higher regret compared to the two-stage approach. In practical experiments, employing early stopping can mitigate overfitting and yield better decisions of SPO than the two-stage method.

The curves for our proposed Inv-PnCO is shown in Fig.7 As is observed, though with much higher prediction loss, SPO is able to outperform the two-stage approach with much lower regret due to the generalization loss in Eq (8) is able to reduce the decision error during distribution shifts that include the surrogate loss function Eq (32). This observation also validates the inherent limitation of generalization models of pure machine learning tasks in addressing the generalization issue of predict-and-optimize as it is unaware of the downstream optimization task. Note that though we used early-stopping, we show the full training curves here where the later epochs may show overfitting.

### C.4.2 QUALITATIVE ANALYSIS

We conduct a qualitative analysis of the knapsack dataset on 20 items. As shown in Fig 8(a), we visualize the values and weights of items, and in Fig 8(b), we visualize the item value predictions for ERM (SPO) and Inv-PncO (SPO). By Fig 8(b), due to the differences between the training and testing distributions, we observe significant discrepancies between the predicted values and the true values. However, through our training with Inv-PnCO, we learned features that are more critical for decision-making. For instance, the predictions for items 0 and 3 are significantly lower, allowing the exclusion of such items with high costs but low real values during the subsequent solving stage. Similarly, items 5, 10, and 17 are excluded compared to the solution obtained by ERM. This approach enables the selection of fewer items while achieving lower regret and better final decisions.

### C.4.3 SENSITIVITY ANALYSIS

We assess the sensitivity of Inv-PnCO framework on the knapsack problem across various optimization parameters, encompassing the constraint (the capacity in the knapsack) and the size of decision variables (number of items), as illustrated in Fig 9(a~b). It is evident that our Inv-PnCO framework consistently reduces regret in comparison to ERM across diverse optimization parameters. To quantify this improvement, we employ relative regret, defined as the ratio of regret relative to the full optimal objective given the variability in optimal objectives across different configurations.

Furthermore, we investigate the sensitivity in Fig. 9(c~d) concerning training hyperparameters, specifically the number of environments and the hyperparameter $\beta$. Notably in Fig. 9(c), on the knapsack problem with the default setting, our model exhibits stability across varying values of

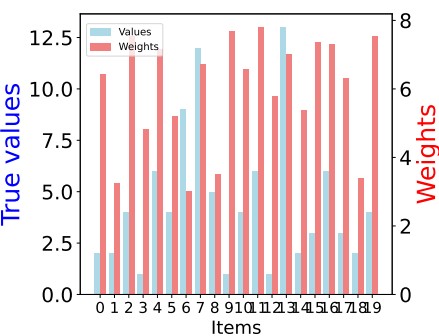
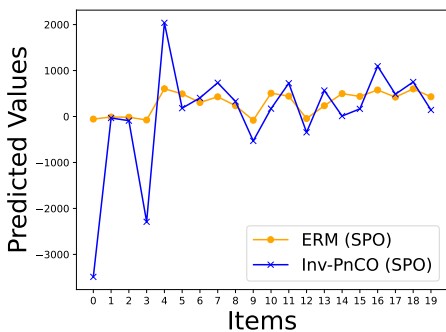

(a) Values & weights of the 70th knapsack instance (b) Predicted values of ERM(SPO) and Inv-PnCO(SPO)

Figure 8: Qualitative analysis on knapsack dataset, where Inv-PnCO improves final decision quality by learning decision-oriented features. (b) visualizes the predicted values for items, and Inv-PnCO demonstrates more appropriate predictions that lead to better final decisions. The selected items for ERM is $\{5, 6, 10, 14, 17, 18\}$, and for Inv-PnCO is $\{4, 6, 7, 16, 18\}$. Inv-PnCO achieves lower regret (of 10) than regret (of 21) in ERM with fewer selected items.

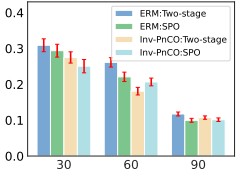
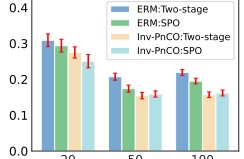
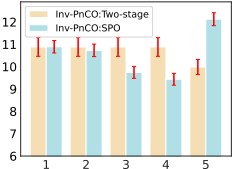
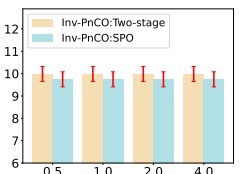

(a) Constraint (capacity)    (b) Decision variable size    (c) Num of environments    (d) Hyper-parameter $\beta$

Figure 9: Sensitivity analysis in regret of Inv-PnCO on knapsack problem w.r.t. optimization parameters (constraint, decision size), and training hyper-parameters (number of environments, $\beta$).

$\beta \in \{0.5, 1.0, 2.0, 4.0\}$. When $\beta$ is too large, it may cause instability in training or amplify the impact of some spurious features. If $\beta$ is too small, it may fit the average of multiple environments. In Figure 3(d), the fluctuation is not very obvious, possibly because the range of beta we chose is not wide enough, but the proper selection of $\beta$ is indeed an important issue. Besides, in Fig. 9(d), we identify that the performance of Inv-PnCO improves at the beginning, then degrades along with the increasing number of environments and achieves its lowest regret when the number of environments is set as 4.

**Parameter sensitivity for optimization problems** We evaluate parameter sensitivity for optimization problems in Fig 9(a~b) for the knapsack under certainty. Fig 9(a) illustrates the results under various constraints (while other parameters are kept unchanged). Fig 9(b) illustrates the results under increasing decision variable size among 20, 50, and 100 (i.e., number of items for the knapsack), where the capacity constraints change proportionally (60, 150, 300) along with the number of variables. Due to the change of the optimal objective along with parameters (constraints or decision variable size), we represent the y-axis results as the ratio between regret and optimal value.

**Hyper-parameter sensitivity in training** We evaluate parameter sensitivity for optimization problems in Fig 9(c~d), while other parameters are set as default as in Table 3. Fig 9(c) illustrates results in regret with respect to hyper-parameter $\beta$. Fig 9(d) illustrates results in regrets with respect to the number of environments during training of Inv-PnCO.

### C.4.4 ABLATION STUDY

We show the results of the ablation study in Table 7. If the variance term is omitted and optimization of the mean term in the loss is solely conducted through the acquired environment, the performance may decline with higher regret, potentially with higher regret than the ERM method directly trained on the train distribution. This validates the necessity of using the regularization term (the variance term in Inv-PnCO loss) to ensure invariant ability for robust decisions.

Table 7: Ablation Study on 3 optimization tasks under uncertainty. Performance degrades if Inv-PnCO is trained without the variance term.

| | Knapsack | | Shortest Path | | Traveling Salesman Problem | |
| --- | --- | --- | --- | --- | --- | --- |
| | Two-stage | SPO | Two-stage | SPO | Two-stage | SPO |
| ERM | 11.22000 | 10.67000 | 18.73675 | 13.68741 | 143.32407 | 104.42732 |
| Inv-PnCO | 9.98500 | 9.10000 | 13.5696 | 13.04145 | 100.50798 | 100.35209 |
| Inv-PnCO w/o Var | 13.69500 | 12.66500 | 46.85968 | 78.45152 | 129.90215 | 136.49825 |

## D    LIMITATIONS

Our approach is based on the core assumption, Assumption 1, which posits that invariant features exist that directly determine the final solution, while spurious features are entirely generated by the environment. However, if this assumption is not satisfied, it may adversely affect the performance of our proposed method.

One potential limitation of this work is that we assume the access to the ground coefficients $\mathbf{y}$ to evaluate decision quality by regret following the previous literature [42; 66; 22; 43]. Regret may not be applicable to evaluate optimization under uncertainty if the ground coefficients $\mathbf{y}$ are unknown for some CO problems. However, in our conducted experiments, $\mathbf{y}$ is available, and this assumption could be satisfied.

In our experiments, solving larger-scale optimization problems may be a future direction. The scale of optimization problems is a major bottleneck for existing predict-and-optimize methods, and our Inv-PnCO approach, based on these PnO methods, will also face scalability issues.

We also assume the parameters in constraints are known and fixed following the literature in predict-and-optimize [42; 16; 62; 43]. As we notice that a few works [28; 29] have been proposed to tackle predict-and-optimize with uncertain constraints, we leave the generalizability exploration of such problems as future work.

We also assume access to diverse training environments during training following previous literature [37]. Future works may involve devising models that mitigate reliance on accessible environments.

## E    BROADER IMPACTS

In our assessment, we have not discerned serious adverse social implications arising from this study. We hope that more robust predict-and-optimize models proposed in our work could be useful to mitigate the risks of decision-making faced by individuals, enterprises, and institutions in uncertain combinatorial optimization problems, thereby reducing the real-world losses associated with the degradation of decision quality of distribution shifts. We acknowledge that this tool may occasionally exhibit suboptimal decision-making quality that is not as good as anticipated during real-world deployments, particularly when there is a huge distribution shift on CO instances or there is not a sufficiently diverse environment to train Inv-PnCO to its best performance. This could potentially lead to losses induction processes. However, it is important to note that this tool is not designed as a general-purpose tool for public use. Moreover, the decisions made by this tool serve merely as decision recommendations, with the ultimate decision-making authority resting with the tool's users. Therefore, it is unlikely to have a widespread or significant negative social impact.

