# OpenReview forum: "Invariant Predict-and-Combinatorial Optimization under Distribution Shifts"
_ICLR.cc/2026/Conference — Submitted to ICLR 2026_

### Official Review · Reviewer_fRtK · 2025-10-18

**Soundness:** 3
**Presentation:** 3
**Contribution:** 3
**Rating:** 6
**Confidence:** 3

**Summary:**

This paper addresses the challenge of out-of-distribution (OOD) generalization in the predict-and-optimize paradigm for combinatorial optimization (CO). The authors observe that when the distribution of input instances shifts between training and testing, the quality of decisions (solutions) can deteriorate significantly. To mitigate this, they propose an Invariant Predict-and-Combinatorial Optimization (Inv-PnCO) framework that augments the training objective with a term to minimize the divergence between the model’s solution distribution and the true (optimal) solution distribution, while adding a regularizer to capture invariant decision-related features across environments. They provide a theoretical analysis (Theorem 1) showing that the proposed objective can reduce OOD decision error, under an invariance assumption. Empirically, Inv-PnCO is evaluated on three CO tasks (knapsack, visual shortest path, and traveling salesman) under different types of distribution shifts (e.g. artificial, perceptual, topological) and consistently improves solution quality compared to standard training

**Strengths:**

- Important Problem: Tackles the under-explored yet practically important problem of OOD generalization in predict-and-optimize pipelines, identifying why standard methods fail to maintain solution quality under shifts.

- Novel Framework with Theory: Proposes a new training framework (Inv-PnCO) with a principled objective that aligns predicted solution distributions with true solutions and enforces invariant features. This approach is backed by a theoretical guarantee (Theorem 1) on reducing OOD error, lending credence to its soundness.

- Clarity and Context: The paper is generally clear and well-structured. It provides relevant context by comparing to previous work (Table 1) and uses examples (Fig. 1) and definitions to build intuition, which helps the reader understand the motivation and significance of the approach.

**Weaknesses:**

- Strong Invariance Assumption: The method relies on Assumption 1, that there exist invariant decision-related features across environments that fully determine the optimal decisions. If this assumption is violated (i.e. if some environment-specific factors also influence the decisions), the theoretical guarantees may not hold and the performance of Inv-PnCO could degrade. The paper would benefit from discussing the realism and limits of this assumption.

- Need for Environment Partitioning: Inv-PnCO’s training procedure appears to require data from multiple environments or an environment label to enforce invariance. It is unclear how the approach would work if environment identifiers are not given or obvious. In practice, obtaining a diverse set of training environments or knowing when a distribution shift occurs can be challenging, which may limit the direct applicability of the method.

- Baseline Comparisons: The experimental comparison is primarily against a vanilla ERM (standard training) baseline. No specialized domain generalization or robust optimization baselines (e.g., IRM, DRO, or other invariant feature learning methods) are evaluated, even though some could potentially be adapted to the predict-and-optimize context. This leaves some uncertainty about how much improvement Inv-PnCO offers over best possible alternatives; a stronger empirical comparison would solidify the claims.

- Complexity and Overhead: The proposed framework introduces additional complexity in training. There are new hyperparameters (such as the regularization weight β and the number of environment splits) and the method incurs higher training cost (e.g. the end-to-end SPO model within Inv-PnCO roughly doubles or more the training time in experiments). While not prohibitive in the tested scenarios, this overhead could hinder scaling to very large or time-sensitive CO problems. The authors do provide some ablation and sensitivity analysis, but the paper could better discuss the trade-off between robustness gains and computational cost.

**Questions:**

- Environment Knowledge: Does the approach require explicit environment labels or a known partition of training data by environment? If so, how would Inv-PnCO perform when such labels are unavailable or when encountering a completely novel type of distribution shift at test time?

- Assumption Validity: How realistic is Assumption 1 in typical applications? If some features that influence decisions do vary across environments (i.e. only partially invariant factors exist), can the method still learn useful invariances, or would it be misled by spurious correlations?

- Alternative Baselines: Could the authors clarify why they did not compare against existing domain generalization techniques (e.g., invariant risk minimization or distributionally robust optimization adapted to the prediction stage)? This would help in understanding whether the improvements stem from the specific Inv-PnCO loss or from generally using multiple environments in training.

- Hyperparameter Sensitivity: How sensitive are the results to the choice of the regularization weight β and the number of environment splits used during training? For instance, is there a risk of under or over-regularization affecting the predictive accuracy versus the decision quality trade-off? Any guidelines from the sensitivity analysis would be useful to know.

- Residual OOD Gap: Even with Inv-PnCO, there remains some gap between in-distribution and OOD performance (the OOD regret, while improved, is still higher than IID in the experiments). What are the remaining sources of error under distribution shift, and could further techniqueshelp close this gap?

---

> ### Author Response · Authors · 2025-11-20
>
> We thank reviewer fRtK for the thorough reviews, constructive suggestions, and insightful questions. We provide detailed answers to your comments below.
>
> > **Weakness 1:** Strong Invariance Assumption & **Question 2:** Assumption Validity
>
> While we agree that for Assumption 1, its practical verification is challenging. Tt is a necessary abstraction for providing a principled approach to handling distribution shifts in combinatorial optimization (CO) problems. The concept of invariant features is commonly leveraged in the broader area of out-of-distribution (OOD) generalization (e.g., domain adaptation, transfer learning). Similar assumptions of invariance under environment shifts have been used in many OOD generalization works. Our paper seeks to build on this tradition while extending it to combinatorial optimization problems.
>
>
> > **Weakness 2:** Need for Environment Partitioning
>
> Access to multiple training environments is a common assumption in the out-of-distribution (OOD) field. In practice, since we do not assume prior knowledge of the test environment, this requirement is feasible to implement, and our experimental results demonstrate the method’s empirical effectiveness.
>
>
> > **Weakness 3:** Baseline Comparisons & **Question 3:** Alternative Baselines
>
> We sincerely appreciate this constructive suggestion. As detailed in Appendix A.2, our analysis extensively engages with existing OOD-related methodologies. However, as highlighted in Table 1, a critical distinction emerges: conventional distribution shift techniques predominantly target **predictive generalization** through standard loss metrics, whereas our framework uniquely addresses **decision-aware generalization** under distribution shifts.
>
> This paradigm shift arises from two inherent characteristics of predict-and-optimize systems:
> - Performance is fundamentally measured through **decision regret** rather than prediction errors, requiring end-to-end sensitivity analysis across both the prediction model and optimization mapping.
> - The coupled learning dynamics necessitate propagating uncertainty through the combinatorial optimization layer, contrasting with the decoupled prediction-focused in baseline methods
>
> Consequently, generalization techniques designed for pure prediction tasks remain orthogonal to our framework, as our theoretical guarantees stem from regret-bound propagation rather than traditional generalization error analysis. We think that alternative methods beyond ERM require significant modifications to the predict-and-optimize models, which justifies our exclusion of additional baselines.
>
>
> > **Weakness 4:** Complexity and Overhead
>
> On scalability, we wish to emphasize that scalability presents a critical challenge for current predict-and-optimize methodologies (including the SPO baseline in our experiments). Existing approaches often face scalability bottlenecks when handling large-scale CO problems, particularly in terms of computational efficiency. While our method’s runtime performance remains inherently tied to the underlying optimization solver, we observe that conventional SPO implementations incur significant computational overhead during training phases.
>
> > **Question 1:** Environment Knowledge
>
> The approach does not require explict knowledge on the eenvironment labels. In theory, as the number of environments increase, our approach is easier to generalize.
>
> > **Question 4:** Hyperparameter Sensitivity
>
> The sensitivity of hyperparameters of  β and the number of environments are shown in Figure 9(c~d) of Appendix C.4.3.
> The selection of number of environments rely on the trade-off of computational cost and generalization ability. For hyperparameter β, theoretically, when β is too large, it may cause instability in training or amplify the impact of some spurious features. If β is too small, it may fit the average of multiple environments. In practice, the variance is not large when β changes.
>
> > **Question 5:** Residual OOD Gap
>
> While a performance gap between IID and OOD results persists, we identify two key remaining sources of this gap: (1) Limitations of  generalization implementation: We design an approximate objective to achieve generalization, but practical constraints—such as computational limits—prevent access to infinitely many training environments. (2) Limitations of generalization assumptions: When the assumptions (e.g., the invariance assumption) are violated, the method cannot fully realize its generalization potential. However，we note any design of genearlization methods cannot exist without assumptions, and the gap can be reduced but may not be removed.

---

> > ### Comment · Reviewer_fRtK · 2025-11-21
> >
> > Thank you for the detailed rebuttal and for addressing the points I raised. I appreciate the clarifications regarding the invariance assumption, environment construction, and the baseline choices. While your responses were helpful, some of my core concerns—particularly regarding the practicality of the invariance assumption and the absence of stronger adapted baselines—remain only partially resolved. For this reason, I will keep my original score.

---

### Official Review · Reviewer_icwq · 2025-10-29

**Soundness:** 3
**Presentation:** 1
**Contribution:** 2
**Rating:** 4
**Confidence:** 2

**Summary:**

The paper focuses on the predict-and-optimize approach in the combinatorial optimization setting. Specifically, the authors consider the out-situation where the the distribution can shift over time. To this end, the authors introduced the inv-PnCO method. This method uses regularization to find invariant features across environments. To illustrate the model's effectiveness, the authors derive a surrogate for the loss, and experimental results on various optimization problems.

**Strengths:**

-	The experiments include optimization problems of different types. Also, the results show on average, an improvement is made using the proposed new method
-	The contribution of extending the predict-and-optimize to the combinatorial optimization setting is relevant

**Weaknesses:**

-	The discussion is very brief. Although the authors provide more extensive limitations in the appendix, I would argue that it is important to include a more elaborate discussion in the main text
-	Code of experiments is not provided as supplementary material or as an anonymized GitHub. This raises concerns regarding reproducibility
-	Figure 3a has Risk on the y-axis, but there is no scale / are no numbers. Besides, Risk is not defined
-	Minor: There are punctuation errors. In some equations, e.g., eq. 8, commas or stops are missing, and/or the next new line has a mismatched use of capitals.

**Questions:**

-	The method is introduced for the Combinatorial Optimization setting. It seems this method is still applicable to the non-combinatorial setting. Could the authors elaborate whether this is correct?
-	The assumption used seems strong. Could the authors elaborate on which situations or contexts the assumption holds and when it breaks?
-	The results show in the tables show an improvement. However, is this number the average over test-points? How large are standard deviations?

---

> ### Author Response · Authors · 2025-11-20
>
> We thank reviewer icwq for the thorough reviews, constructive suggestions, and insightful questions. We provide detailed answers to your comments below.
>
> > **Weakness 1:**  The discussion is very brief. Although the authors provide more extensive limitations in the appendix, I would argue that it is important to include a more elaborate discussion in the main text
>
> Thanks for the nice suggestion, and within the page limit, we would try to include more in the main text.
>
>
> > **Weakness 2:**  Code of experiments is not provided as supplementary material or as an anonymized GitHub. This raises concerns regarding reproducibility.
>
> Thanks for your suggestion. Our code will be released after publication.
>
>
> > **Weakness 3:**  Figure 3a has Risk on the y-axis, but there is no scale / are no numbers. Besides, Risk is not defined
>
>
> Thanks for the nice suggestion, we will revise Figure 3(a). Risk is adopted from the field of out-of-distribution genearlizaton, and is the training loss of our model.
>
> > **Weakness Minor:**: There are punctuation errors. In some equations, e.g., eq. 8, commas or stops are missing, and/or the next new line has a mismatched use of capitals.
>
> Thanks for pointing out the typos, and we will revise accordingly.
>
> > **Question 1:** The method is introduced for the Combinatorial Optimization setting. It seems this method is still applicable to the non-combinatorial setting. Could the authors elaborate whether this is correct?
>
> This is correct that this approach applies to non-combinatorial setting, since the main theoretical results (Theorem 1 and Proposition 1) remain unchanged when the problems are extended to combinatorial optimizaton problems.
>
> > **Question 2:** The assumption used seems strong. Could the authors elaborate on which situations or contexts the assumption holds and when it breaks?
>
> As illustrated in the Introduction and Figure 1 of the main paper, the assumption holds in scenarios where core structural properties remain invariant to distribution shifts—for example, the shortest path in a network stays unchanged despite variations in perceptual mechanisms. Regarding cases where the invariance assumption breaks, a comprehensive enumeration is challenging: many invariant variables are latent (rather than explicit), making it difficult to definitively demonstrate the absence of such invariances.
>
> > **Question 3:** The results show in the tables show an improvement. However, is this number the average over test-points? How large are standard deviations?
>
> The results shows the average over test points.
>
> Unlike machine learning prediction problems, in optimization problems, the standard deviation (std) of a single solver call is related to the distribution of the samples. Since we did not perform multiple runs, there is no std available for direct evaluation.

---

### Official Review · Reviewer_ErZU · 2025-10-31

**Soundness:** 2
**Presentation:** 2
**Contribution:** 2
**Rating:** 4
**Confidence:** 4

**Summary:**

This paper proposes Invariant Predict-and-Combinatorial Optimization (Inv-PnCO), a framework for decision-focused learning that is robust to distribution shifts.  Their approach adapts loss functions from decision-focused learning by adding a regularization term.  Empirically, their approach outperforms the baselines on the knapsack, shortest path, and traveling salesman problems.

**Strengths:**

- **Motivation**: Overall, this is definitely a well-motivated extension of the predict-and-optimize line of research.
- **Reported Results**:  Although there are some issues with reporting, as I will highlight in weaknesses, the overall averaged results demonstrate improvement over ERM in terms of decision quality.

**Weaknesses:**

- **Scalability**: The proposed approach is not computationally efficient, with train/test times often being significantly higher than ERM with SPO.  This, combined with the relatively small size of the instances, signals a limited scalability.
- **Distribution Shift Results and Reporting**: Ultimately, given that the focus of the paper is on distribution shifts, more needs to be done with respect to this.  The authors explore a single type of shift per problem and aggregate them across all different environments.  While this provides some information, de-aggregating and providing reporting of how each level of distribution shift is needed. Moreover, analysis beyond average regret, e.g., distributional/CVaR/worst-case, would give much more information in evaluation as well.

  Building on this, presenting the experimental setup and comprehensive results for the distribution in the main paper would be appreciated.  Currently, only limited results are reported.  However, including shift-specific definitions and parameters, as well as results, should be prioritized in the main paper over redefining the standard predict-and-optimize approach.  One suggestion for the main paper results would be to use boxplots that demonstrate the distributional performance in each environment, rather than relying on single metrics averaged across all environments.

- **Clarity**: One major weakness is the writing and clarity of the paper. Significant aspects, such as what the authors mean by 'environment,' are not clearly specified when introduced (an example would be helpful), ultimately making readability an issue.

- **Author Engagement with Revisions**: I have reviewed this paper in the past, and I believe my colleagues have as well.  The authors have not made any notable changes to the manuscript after several resubmissions, so I am hesitant to recommend acceptance until I see a clear effort to address the numerous concerns that have not been addressed between resubmissions.

**Questions:**

- Figure 8a x-axis needs to be spaced better.
- How were the environments selected? Have the authors considered a combination of multiple shifts?
- How many instances per environment are evaluated?
- How is the performance of all methods affected by the magnitude of the shift present in each environment?
- Is it possible to improve scalability by leveraging methods that have also been used in speed-up standard SPO frameworks?
- How does the validation set selected affect the quality of the performance under shift?

---

> ### Author Response · Authors · 2025-11-20
>
> We thank reviewer ErZU for the thorough reviews, constructive suggestions, and insightful questions. We provide detailed answers to your comments below.
>
> > **Weakness 1:** Scalability: The proposed approach is not computationally efficient, with train/test times often being significantly higher than ERM with SPO. This, combined with the relatively small size of the instances, signals a limited scalability.
>
> Thanks for the nice suggestion, and we would try our best to give results with scalable settings. On scalability, we wish to emphasize that scalability presents a critical challenge for current predict-and-optimize methodologies (including the SPO baseline in our experiments). Existing approaches often face scalability bottlenecks when handling large-scale CO problems, particularly in terms of computational efficiency. While our method’s runtime performance remains inherently tied to the underlying optimization solver, we observe that conventional SPO implementations incur significant computational overhead during training phases.
>
> > **Weakness 2:** Distribution Shift Results and Reporting
>
> Thanks for the great suggestions, and we would consider re-organize the paper to better illustrate results.
>
> > **Weakness 3:** Clarity: One major weakness is the writing and clarity of the paper. Significant aspects, such as what the authors mean by 'environment,' are not clearly specified when introduced (an example would be helpful), ultimately making readability an issue.
>
> For the environment, we note that "environment" is also an abstract term used in this context to quantify distribution shifts, and invariant factors & spurious factors (where we do not introduce spurious factors in this paper) are latent factors that quantify the influence of distribution shifts on the final output. Therefore, the main purpose of introducing these terms and necessary assumption is to guide the model design toward greater generalizability. We will introduce more in the revision.
>
> > **Weakness 4：** Engagement with revisions
>
> We sincerely thank the reviewer for their ongoing feedback. We acknowledge that prior revisions did not fully address all concerns and take responsibility for this.  We hope the further version would address feedback comprehensively.
>
>
> > **Question 1:** Figure 8a x-axis needs to be spaced better.
>
> Thanks for the nice suggestion and we will revise accordingly.
>
> > **Question 2:** How were the environments selected? Have the authors considered a combination of multiple shifts?
>
> Environments are selected with distict hand-crafted data distributions. For now combination of multiple shifts are not used.
>
> > **Question 3:** How many instances per environment are evaluated?
>
> The statistics are listed below:
>
>  Problem                                 |  Train samples |  Test samples |
> | :--------------------------------------- | :---------------: | :-------------- |
> | Knapsack (Probabilistic shift Knapsack) | 400             | 200            |
> | Shortest path(Perceptual shift)         | 10000           | 1000           |
> | TSP (Topological shift)                 | 400             | 200            |
>
>
> > **Question 4:** How is the performance of all methods affected by the magnitude of the shift present in each environment?
>
> While we did not conduct dedicated experiments varying the magnitude of distribution shifts directly, Figure 9(a) offers relevant insights: as constraints are tightened, the observed performance trends align with those expected under larger shift magnitudes.
>
> > **Question 5:** Is it possible to improve scalability by leveraging methods that have also been used in speed-up standard SPO frameworks?
>
> Some approaches [1,2] in SPO framework use to improve inference time through model-based optimization objective estimation, however the time in pretraining stage cannot be saved.
>
> > **Question 6:** How does the validation set selected affect the quality of the performance under shift?
>
> Though we do not conduct experiments due to time limit, as the training environments are sufficient, the selection of validation would not affect much.
>
>
>
> [1] Shah, Sanket, et al. "Decision-focused learning without decision-making: Learning locally optimized decision losses." Advances in Neural Information Processing Systems 35 (2022): 1320-1332.
> [2] Zharmagambetov, Arman, et al. "Landscape surrogate: Learning decision losses for mathematical optimization under partial information." Advances in Neural Information Processing Systems 36 (2023): 27332-27350.

---

### Official Review · Reviewer_p4ny · 2025-11-01

**Soundness:** 3
**Presentation:** 3
**Contribution:** 3
**Rating:** 6
**Confidence:** 2

**Summary:**

The paper studies out-of-distribution generalization for predict-and-optimize (PnO) with combinatorial objectives (knapsack, visual shortest path, TSP). It proposes Inv-PnCO, a plug-in training objective that combines a mean-loss term with a cross-environment variance penalty. The goal is to learn invariant decision-oriented factors so that the induced solution distribution q(z∣x) remains close to the optimal p(z∣x) across environments. Theoretical claims include: 1) an information-theoretic bound suggesting that penalizing I(z;e∣y) reduces OOD decision error, and 2) a tractable surrogate showing that a mean+variance loss over environments upper-bounds the ideal objective. Empirically, on OOD shifts, Inv-PnCO reduces regret vs. ERM in both PtO and PnO pipelines, with no added test-time cost.

**Strengths:**

1. The setup relates to real problems (routing/logistics/planning) where costs are uncertain.

2. The experiments span arrays (knapsack), images (visual SP), and graphs (TSP) with distinct shift types, supporting the generality of Inv-PnCO.

3. The mutual information-based regularization, which converts to the variance across the environment, is naturally motivated. And the way of adding a regularization term does not introduce additional inference cost.

**Weaknesses:**

1. Benchmarks remain small (e.g., TSP-20, short grids). It would be great to test on larger instances, where 1) solver noise and suboptimality increase, 2) regret landscapes get spikier, and 2) environment variance explodes, causing the regularization term hard to balance.

2. Results mainly compare Inv-PnCO to ERM (both two-stage and SPO). Could distributionally robust optimization or OOD generalization-related works also be empirically compared?

3. Many tables emphasize mean regret drops without tight confidence intervals, worst-environment performance.

**Questions:**

1. The paper focuses on the fixed feasible set setting. Would it be possible to extend it to variable feasible sets?

---

> ### Author Response · Authors · 2025-11-20
>
> We thank reviewer p4ny for the thorough reviews, constructive suggestions, and insightful questions. We provide detailed answers to your comments below.
>
>
> > **Weakness 1:** It would be great to test on larger instances, where 1) solver noise and suboptimality increase, 2) regret landscapes get spikier, and 2) environment variance explodes, causing the regularization term hard to balance.
>
> Thanks for the nice suggeston, and we will try to give experiment results on these aspects. On scalability, we wish to emphasize that scalability presents a critical challenge for current predict-and-optimize methodologies (including the SPO baseline in our experiments). Existing approaches often face scalability bottlenecks when handling large-scale CO problems, particularly in terms of computational efficiency. While our method’s runtime performance remains inherently tied to the underlying optimization solver, we observe that conventional SPO implementations incur significant computational overhead during training phases.
>
> > **Weakness 2:** Results mainly compare Inv-PnCO to ERM (both two-stage and SPO). Could distributionally robust optimization or OOD generalization-related works also be empirically compared?
>
> We sincerely appreciate this constructive suggestion. As detailed in Appendix A.2, our analysis extensively engages with existing OOD-related methodologies. However, as highlighted in Table 1, a critical distinction emerges: conventional distribution shift techniques predominantly target **predictive generalization** through standard loss metrics ($\mathbf{X} \rightarrow \mathbf{\hat{Y}}$), whereas our framework uniquely addresses **decision-aware generalization** under distribution shifts.
>
> This paradigm shift arises from two inherent characteristics of predict-and-optimize systems:
> - Performance is fundamentally measured through **decision regret** (quantifying optimality gaps) rather than prediction errors, requiring end-to-end sensitivity analysis across both the prediction model ($\mathbf{X} \rightarrow \mathbf{\hat{Y}}$) and optimization mapping ($\hat{\mathbf{Y}} \rightarrow \mathbf{z}(\hat{\mathbf{Y}})$ )
> - The coupled learning dynamics necessitate propagating uncertainty through the combinatorial optimization layer, contrasting with the decoupled prediction-focused adaptation in baseline methods
>
> Consequently, techniques designed for pure prediction tasks remain orthogonal to our framework, as our theoretical guarantees stem from regret-bound propagation rather than traditional generalization error analysis. We think that alternative methods beyond ERM require significant modifications to the predict-and-optimize models, which justifies our exclusion of additional baselines.
>
> > **Weakness 3:**  Many tables emphasize mean regret drops without tight confidence intervals, worst-environment performance.
>
> We conducted one independent run, as is common practice in predict-and-optimize. In practice, model outputs are generally stable because the final step involves optimization solving, which is deterministic. However, we present the error bar in Figure 4 for the individual run.
>
> > **Question:** The paper focuses on the fixed feasible set setting. Would it be possible to extend it to variable feasible sets?
>
> We agree with the reviewer that Variable feasible sets have practical applications in the real world. Our proposed approach are applicable to this setting, however we put it as future work.

---

### Meta-Review · Area_Chair_jG5k · 2026-01-05

**Summary:**

The reviewers commend the paper for tackling an important and timely problem: how to achieve out-of-distribution (OOD) generalization for predict-and-optimize frameworks in combinatorial optimization under uncertainty. The proposed Inv-PnCO method—using a variance-based regularizer across environments—is well motivated by practical needs in applications such as routing or logistics when faced with a distributional shift. Empirical studies on knapsack, shortest path, and TSP under various types of distribution shifts are viewed as strengths and help validate the generality of the approach.

However, there are significant concerns shared by reviewers that limit enthusiasm. First, the scalability of the method is questioned, as experiments are limited to relatively small-sized instances, and training time increases noticeably. Second, the method relies on a strong invariance assumption, and while this is common in related theory, reviewers question its practical validity and the lack of analysis when it is violated. Third, reviewers expected stronger or more direct comparisons to relevant OOD baselines (such as IRM, DRO, or robust optimization methods), rather than only comparing to ERM within predict-and-optimize. Finally, concerns remain about reporting (e.g., confidence intervals, shift-specific/worst-case results) and clarity of definitions (especially regarding "environment"). Despite a constructive author rebuttal, these key issues are not fully resolved.

Considering this paper is not well-written and the problem is not clear-defined, I tend to reject this paper. Nevertheless, this article presents a very good idea, and I recommend submitting it to subsequent conferences after further refinement

**Reviewer Concerns:**

The authors addressed some requests for clarification and acknowledged presentation/reproducibility issues.

Core concerns on scalability, baseline coverage, and the practicality of assumptions still stand.

**Reviewer Scores:**

Scores are likely unchanged for all reviewers after the rebuttal and discussion.

---

### Decision · Program_Chairs · 2026-01-26

Reject